SCIENCE FORUM

# Nanoscape, a data-driven 3D real-time interactive virtual cell environment

**Abstract** Our understanding of cellular and structural biology has reached unprecedented levels of detail, and computer visualisation techniques can be used to create three-dimensional (3D) representations of cells and their environment that are useful in both teaching and research. However, extracting and integrating the relevant scientific data, and then presenting them in an effective way, can pose substantial computational and aesthetic challenges. Here we report how computer artists, experts in computer graphics and cell biologists have collaborated to produce a tool called Nanoscape that allows users to explore and interact with 3D representations of cells and their environment that are both scientifically accurate and visually appealing. We believe that using Nanoscape as an immersive learning application will lead to an improved understanding of the complexities of cellular scales, densities and interactions compared with traditional learning modalities.

**SHEREEN R KADIR, ANDREW LILJA, NICK GUNN, CAMPBELL STRONG, ROWAN T HUGHES, BENJAMIN J BAILEY, JAMES RAE, ROBERT G PARTON AND JOHN MCGHEE\***

## Introduction

Since the invention of the microscope in the 16th century and the subsequent discovery of a world concealed from our bare eyes, advances in experimental science and technology have excelled our understanding of cell biology. Modern imaging techniques were central to unravelling the intricacies of the cellular landscape at a molecular level. X-ray crystallography, NMR spectroscopy and cryo-electron microscopy achieve atomic- or near-atomic resolution of individual proteins and cell architecture, while live-cell imaging enables observation of cellular structures, processes and behaviours in real-time. Further, the use of enhanced fluorescent tags and biosensors has shed considerable light on protein kinetics, interactions and diffusion (*Wollman et al., 2015*; *Goodsell et al., 2020*).

Three-dimensional (3D) visualisation of scientific concepts and experimental data is nowadays a popular tool to provide insight where traditional, two-dimensional graphical illustrations or descriptive text cannot (*Goodsell and Jenkinson, 2018*). The educational benefits are numerous, from clarifying complex or abstract concepts to testing hypotheses and generating new ideas (*Daly et al., 2016*; *Jenkinson and McGill, 2012*; *McClean et al., 2005*; *Kadir et al., 2020*).

However, no single experimental modality is sufficient to elucidate the structure and dynamics of macromolecular assemblies and cellular processes. Therefore, integrative modelling of data from multiple, complementary techniques and scientific disciplines is crucial (*Ornes, 2016*). Scientific journals and bioinformatics databases contain a wealth of such data, but the extraction, interpretation and consideration of the relevant data is challenging (*Conesa and Beck, 2019*). Moreover, many biomedical visualisations do not cite journals or databases, meaning that users do not know if a given visualisation has been informed by empirical data, or if it owes more to

**\*For correspondence:**
john.mcghee@unsw.edu.au

**Competing interests:** The authors declare that no competing interests exist.

**Figure 1.** Relative scales and temporal dynamics of the cell surface components featured in Nanoscape. A 3D model of a breast cancer cell (upper left panel). The region inside the yellow box is shown in more detail in the lower left panel, and region inside the pink box is shown in more detail in the panel on the right. The lower left panel details the components of the extracellular matrix (such as collagen I and proteoglycans), filopodia and macropinocytosis structures (which engulf extracellular material and fluid). The right panel details extracellular vesicles (exosomes), pits in the plasma membrane (caveolae and clathrin coated pits), plasma membrane lipids, surface proteins, components of the extracellular matrix (proteoglycans, collagens I and IV), and a 20 nm nanoparticle. The light red figure in the right panel is 40 nm tall. See *Video 1* for animations of some of these processes. ECM: extracellular matrix.

artistic license (*Goodsell and Johnson, 2007*; *Jantzen et al., 2015*).

Biomedical animators often use 3D computer animation and modelling software popular in the games and entertainment industry, such as Autodesk Maya (https://autodesk.com/maya), SideFX Houdini (https://www.sidefx.com/products/houdini/) and Pixologic Zbrush (http://pixologic.com/features/about-zbrush.php). However, the true complexity of the cellular environments is often deliberately diminished to clarify or emphasise features and mechanisms of interest. It may also be reduced for other reasons, such as technical limitations of computer graphics or time constraints (*Goodsell and Johnson, 2007*).

The last couple of decades have seen an increase in visualisation software tools and accurately scaled, static reconstructions at the molecular level. Notable examples are the HIV-1 virus and *Mycoplasma mycoides* (created with the packing algorithm CellPACK), and a snapshot of a synaptic bouton obtained through integrating a plethora of imaging techniques, quantitative immunoblotting and mass spectrometry (*Johnson et al., 2015*; *Johnson et al., 2014*; *Wilhelm et al., 2014*). Previously, simulations of Brownian or molecular dynamics have been incorporated into 3D atomic resolution models of bacterial cytoplasmic subsections to examine the effect of the interactions, stability and diffusion of proteins in crowded cellular

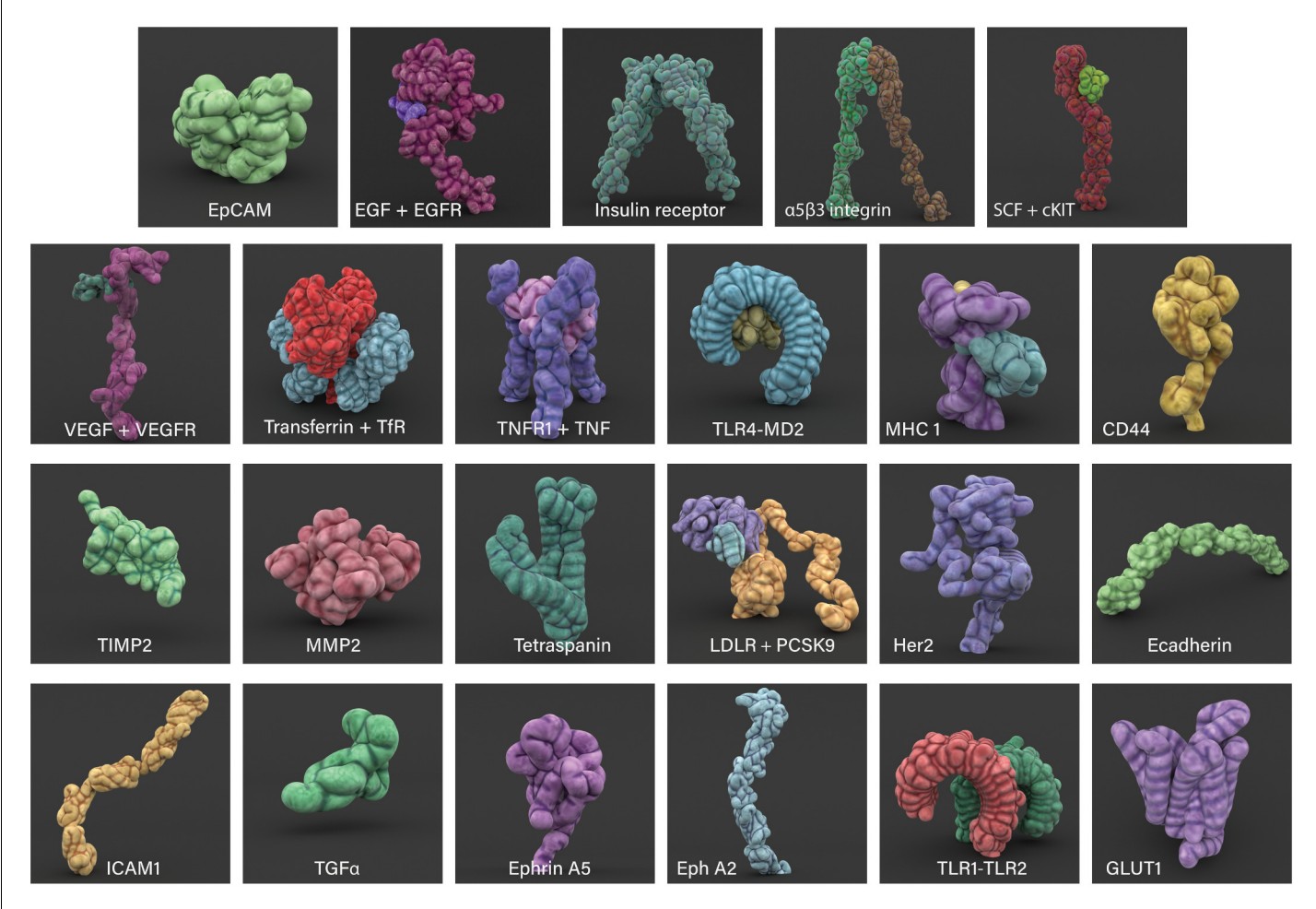

**Figure 2.** 3D modelled cell surface receptors and ligands featured in Nanoscape. Stylized 3D meshes modelled from structures retrieved from the RSCB Protein Data Bank (PDB). Most proteins are depicted as monomers; see Appendix 1 for details.

environments (*Feig et al., 2015*; *Yu et al., 2016*; *McGuffee and Elcock, 2010*).

Furthermore, mathematical and computational modelling platforms such as V-Cell, M-cell, and E-cell are designed to be used by experimental biologists and theoretical biophysicists (*Moraru et al., 2008*; *Stiles and Bartol, 2001*; *Stiles et al., 1996*; *Tomita et al., 1999*). These platforms can run simulations of cell biological phenomena based on a combination of multiple 'omics' technologies and imaging data, thereby enabling scientists to analyse 3D representations of their raw data and test hypotheses with varying degrees of molecular and spatial resolution.

However, many of the existing techniques described above do not fully represent cellular environments, mostly due to sheer complexity but also because of lack of funding or knowledge, or computational limitations. Our previous project (Journey to the Centre of the Cell)

already showed significant improvement in the students' comprehension of cellular structures and processes (*Johnston et al., 2018*). It provided an immersive, virtual reality educational experience of an entire 3D cell based on serial, block-face scanning electron microscope imaging data. Although the project successfully depicted cellular features from real microscopy data, portrayal of the cell surface environment had to be oversimplified due to various constraints, including working at higher virtual reality frame rates (over 90 fps) and a smaller development team.

Nanoscape is a collaborative follow-up project between computer artists, experts in computer graphics and cell biologists to create an interactive real-time open-world experience that enables a user to navigate a cell terrain within a tumour microenvironment. This first-of-its-kind application distils and integrates a vast archive

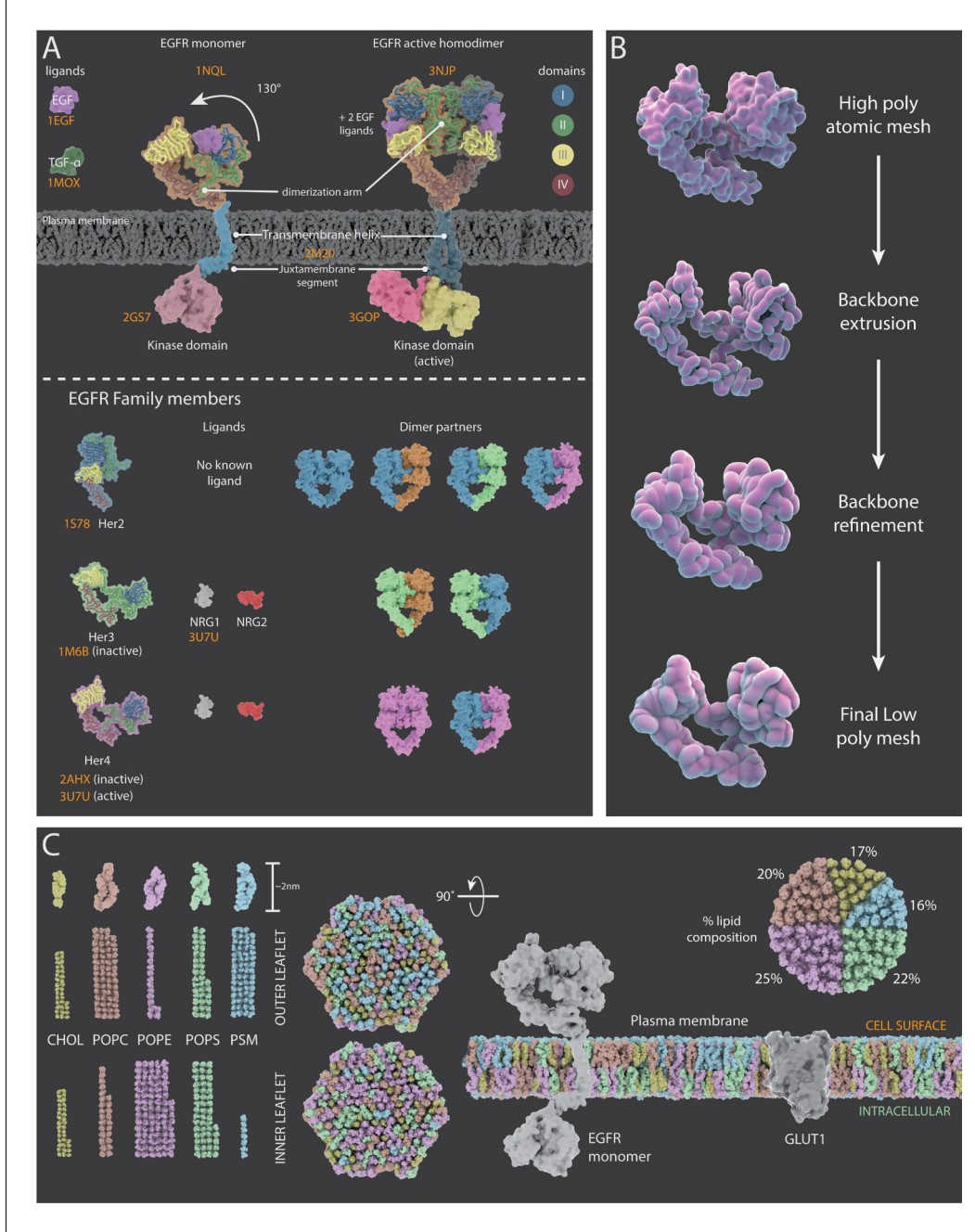

**Figure 3.** Representations of surface receptors and plasma membrane lipids. (**A**) Structural and dynamic information about the four proteins in the ErbB family of proteins (EGFR, Her2, Her3, and Her4) and their ligands. Top: Mechanism of action for EFGR which, upon ligand binding, undergoes a conformational change (130° movement) into the active extended conformation; it can also form a dimer with another active EGFR protein. Bottom: PDB structures, ligands and dimer partner combinations for Her2, Her3 and Her4. (**B**) The creation of stylized protein meshes starts with structures sourced from the PBD (top); the backbone is extruded and the structure is then refined to produce the mesh (bottom). (**C**) A 3D model of a lipid bilayer in a cancer cell, highlighting an asymmetric distribution of 400 lipids (data adapted from *Shahane et al., 2019*). The bilayer components include cholesterol (CHOL), 1-palmitoyl-2-oleoyl-sn-glycero-3-phosphocholine (POPC), 1-palmatoyl-2-oleoyl-sn-glycero-3-phosphoethanolamine (POPE), 1-palmitoyl-2-oleoyl-sn-glycero-3-phospho-L-serine (POPS), and palmitoylsphingomyelin (PSM). The proportion of each lipid species within the outer and inner leaflets is shown on the left; the percentage of each species in the bilayer is shown on the right. The two hexagonal shapes are side views of a model cancer plasma membrane with proteins EGFR and GLUT1.

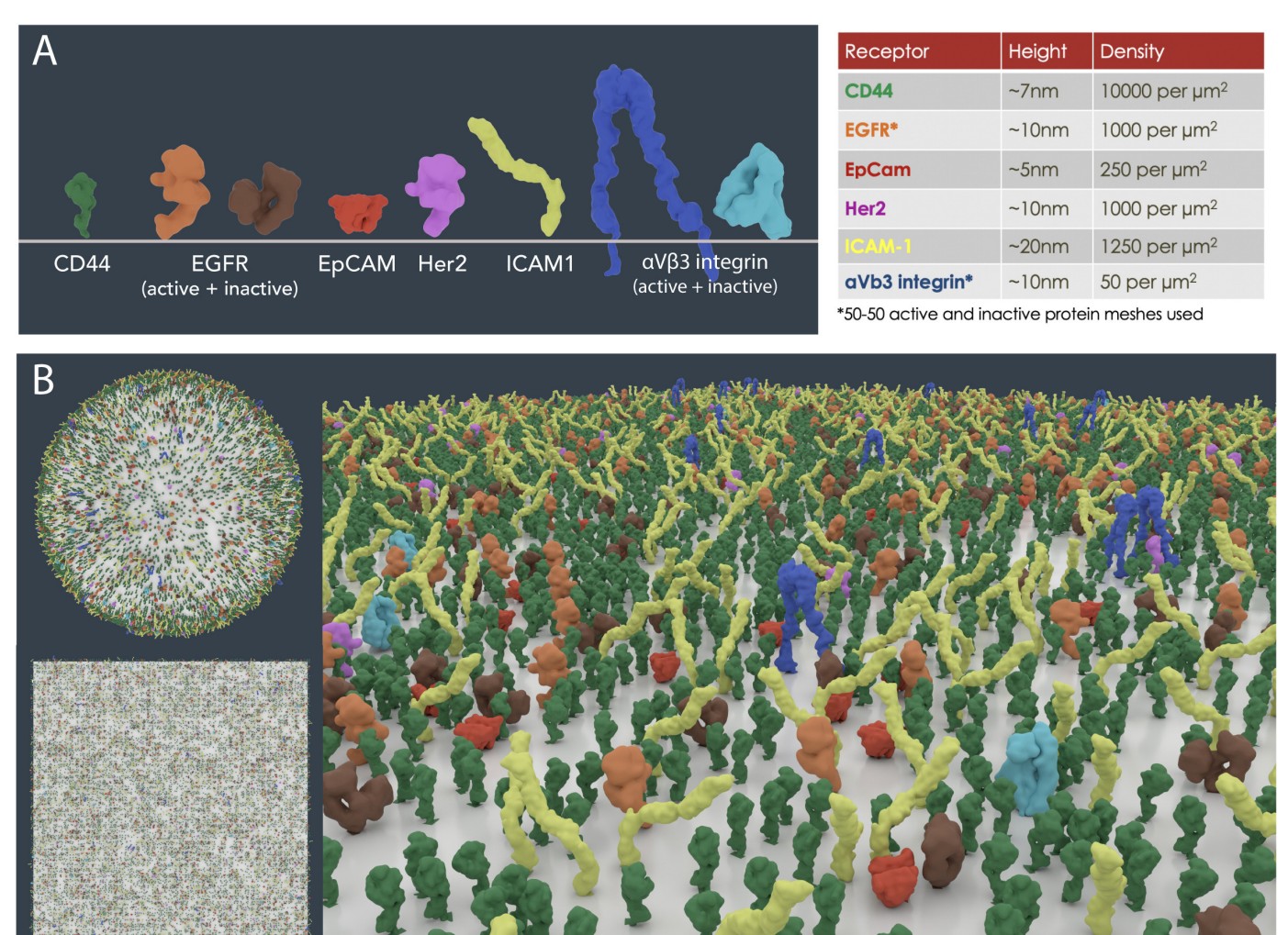

**Figure 4.** Cell surface receptor density modelling from experimental data. (**A**) PDB meshes of 6 well-known surface biomarkers (CD44, EGFR, EpCAM, Her2, ICAM1 and αVβ3 integrin) on MDA-MB-231 cells from flow cytometry data (*Cahall et al., 2015*). (**B**) Scaled receptor meshes were distributed onto a 1 µm² surface area sphere and plane using the autoPACK plugin in Blender.

of scientific data into an interactive immersive experience using a comprehensive library of existing visualisation tools, while employing distinct principles of art and design. Nanoscape is primarily an educational tool for tertiary-level science students, and it is available at https://store.steampowered.com/app/1654050/Nanoscape.

## Using Nanoscape to visualize cell surface proteins and cancer cells

Here, we present some of the challenges and limitations experienced during the data collection process and the conceptualisation of 3D assets, and examine the practicability of visually replicating experimental data. We discuss the implications of gaps in scientific knowledge,

modification or simplification of data, and the use of artistic license for visual clarity. Our work raises important questions about whether molecular visualisations can support outcomes beyond the educational field and could help experimentalists to better understand their data.

As part of the pre-production stage, information on the major surface components, extracellular features and processes commonly found in breast cancer scenarios was collated through a comprehensive review of the literature along with analysis of experimental data obtained from scientific collaborators (*Figure 1*). This cell type was chosen as the focus of the visualisation due to the abundance of protein and structural data available from these sources.

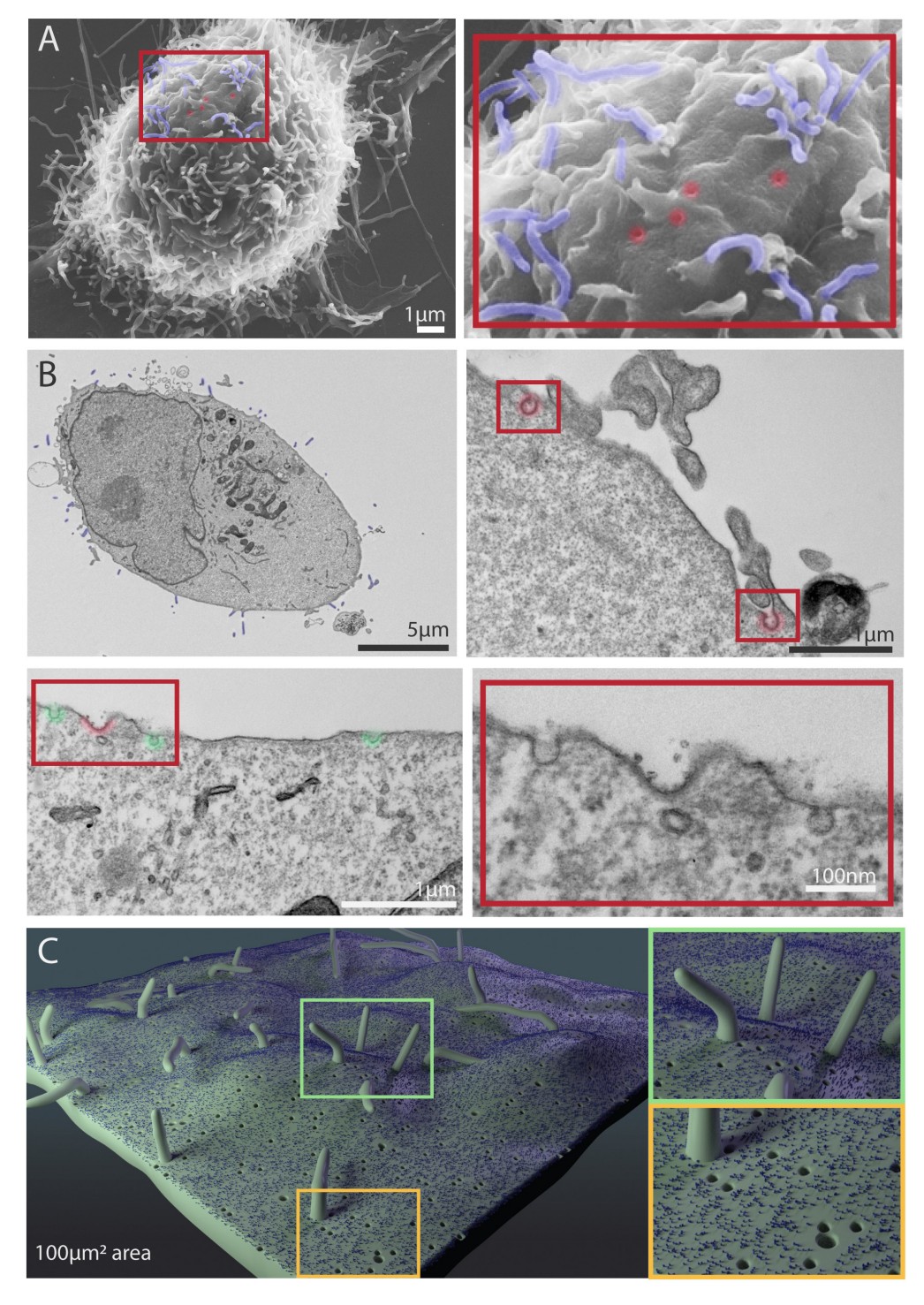

**Figure 5.** Scanning electron microscope (SEM) and transmission electron microscope (TEM) images of MDA-MB-231 cells showing cellular features used in Nanoscape. (A) Representative SEM images of MDA-MB-231 cells. The boxed area shows a higher magnification of filopodia (pseudocoloured blue) and putative pits (caveolae, clathrin coated pits) in red. (B) Representative TEM images of sections of MDA-MB-231 cells. Filopodia are visible in the upper left panel (pseudocoloured blue). Clathrin coated pits are visible in the upper right and lower left panels (red) and right. Caveolae are visible in the lower left panel (green). The lower right panel shows the region inside the red box in the lower left panel at higher magnification. (C) 3D depiction of filopodia, caveolae, clathrin coated pits, and a representative receptor on a 100 μm² patch on the cell membrane.

**Table 1.** Cellular structures and processes in Nanoscape.

| | Feature | Examples | Dimensions | Density | Temporal dynamics |
|---|---|---|---|---|---|
| Structures | Membrane bound receptors[1] | EGFR, Integrins, VEGFR | ~5–20 nm length; ~1–5 nm diameter | Protein specific ~10–10$^4$ per µm$^2$ | Protein specific D ~ 10$^{-3}$–1 µm$^2$/s [see note 10]; Transitions between protein states ~ 1–100µs |
| | Soluble proteins[1] | EGF, TNF, MMPs | | | |
| | Membrane transport proteins[2] | GLUT4, K$^+$ channel, Na$^+$/K$^+$ pump | ~4–10 nm length | ~10$^4$ per µm$^2$ | ~10$^0$–10$^7$/s transport rate |
| | Extracellular matrix[3] | Collagens, Fibronectin, Hyaluronic acid | Variable: from large fibres to smaller glycoproteins | Variable | Very low mobility relative to proteins |
| | Plasma membrane[4] | Phospholipids | ~2 nm length;~0.25–0.5 nm$^2$ cross-sectional area | ~5×10$^6$ per µm$^2$ | $^†$D ~ 1 µm$^2$/s |
| Processes | Protrusions | Filopodia[5] | ~1–5 µm length; ~150–200 nm diameter | ~0.3 per µm$^2$ | ~25–50 nm/s protrusion rate |
| | Endocytosis | Caveolae[6] | ~65 nm mean diameter; ~0.0067 µm$^2$ area | ~10 per µm$^2$ | ~30 s to minutes |
| | | Clathrin mediated endocytosis[7] | ~110 nm mean diameter; ~0.0190 µm$^2$ area | ~0.8 per µm$^2$ | ~30–60 s |
| | | Macropinocytosis[8] | ~0.2–5 µm diameter | ? | ~120 s |
| | Extracellular vesicles | Exosomes[9] | ~40–150 nm diameter | ? | ? |

\* Key features of cellular structures and processes in Nanoscape, with examples detailing properties such as dimensions, densities, temporal dynamics. See *Figure 1* for 3D models.

Notes: 1 Membrane bound receptors and soluble proteins. **Milo et al., 2010**. 2 Membrane transport proteins. **Milo et al., 2010**. Chapter IV in **Milo and Phillips, 2015**. Page 9 (top paragraph) in **Itzhak et al., 2016**. Table 8.3 in **Gennis, 1989**. 3 Extracellular matrix. **Frantz et al., 2010**. **Insua-Rodríguez and Oskarsson, 2016**; **Früh et al., 2015**; **Mouw et al., 2014**; **Pankov and Yamada, 2002**. 4 Plasma membrane lipids. **Milo et al., 2010**. Chapter II in **Milo and Phillips, 2015**. Chapter 10 in **Alberts et al., 2002**. Table 1 in **Rawicz et al., 2000**. Page 2644 (right column, 2nd paragraph) in **Brügger et al., 2006**. 5 Filopodia density and dimensions. Measured from scanning electron micrographs, see Figure 5 in **Mallavarapu and Mitchison, 1999**. 6 Caveolae density, dimensions and temporal dynamics. **Parton, 1994**; **Parton et al., 2020a**; **Parton et al., 2020b**; **Pelkmans and Zerial, 2005**; **Boucrot et al., 2011**; **Richter et al., 2008**. 7 Clathrin mediated endocytosis density, dimensions and temporal dynamics. **Cocucci et al., 2012**; **Doherty and McMahon, 2009**; **Edeling et al., 2006**; **Kirchhausen, 2009**; **McMahon and Boucrot, 2011**; **Merrifield et al., 2002**; **Parton, 1994**; **Saffarian and Kirchhausen, 2008**; **Taylor et al., 2011**. 8 Macropinocytosis dimensions. **Condon et al., 2018**; **Lim and Gleeson, 2011**. 9 Exosome dimensions. **Skotland et al., 2017**. 10 Diffusion coefficient. D is microscopically determined by the velocity of the molecule and the mean time between collisions.

### Cell surface proteins

Cell surface proteins, collectively known as the surfaceome, exhibit a wide range of roles, from playing a vital role in communication between the cell and its environment to signal transduction and transport of ions and other small molecules (**Bausch-Fluck et al., 2018**). The Cell Surface Protein Atlas (wlab.ethz.ch/cspa) and the in silico human surfaceome (wlab.ethz.ch/surfaceome), which together have classified 2886 entries, were used to first identify different types of surface proteins, such as receptors, soluble and membrane transport proteins (**Bausch-Fluck et al., 2015**; **Bausch-Fluck et al., 2018**). Subsequently, over 30 prevalent surface proteins associated with breast cancer that were available from the RSCB Protein Data Bank (PDB) were selected and organised in a cast of characters (see Appendix 1 and *Figure 2*; **Berman et al., 2000**). Where possible, any available information on their motion (molecular dynamics,

conformational changes and protein interactions), and population densities were interpreted from various published sources.

The structural and dynamic information gathered about the ErbB family of proteins (which consists of 4 receptor tyrosine kinases: EGFR, Her2, Her3, and Her4) and the associated literature references used to create mechanism of action animations for each family member are summarised in *Figure 3* and Appendix 2. Molecular Maya (mMaya) modelling and rigging kits (https://clarafi.com/tools/mmaya/), which qualitatively replicate molecular dynamics, were used to simulate ligand binding events and transitions between conformational states (see Materials and methods). These rudimentary mechanism of action animations provided an interpretation of protein movement based on experimental data available and were subsequently used to inform the artistic design team on how best to approach rigging a refined, stylised 3D protein

mesh using traditional rigging methods (*Figure 3B*).

Conformational flexibility plays a crucial role in enabling protein-ligand interactions, multi-specificity and allosteric responses. Unfortunately, most proteins have no reliable, or only partial experimentally determined 3D structures, available. Such limitations may lead to a presentation of a single, 'native' structure in visualisations, instead of multiple flexible conformational variations, further perpetuating misconceptions about protein structure, folding, stability and effects of mutations (*Robic, 2010*). Where possible, Nanoscape used several conformational states of proteins sourced from the PDB as well as conformational changes simulated with mMaya (see Materials and methods). The choice of protein mesh detail and representation will also affect the viewer. A low poly mesh will significantly reduce the computational burden in rendering but may compromise important scientific information, such as the specificity of a ligand binding pocket that may be essential for conveying the mechanism of action. Nanoscape incorporates a level-of-detail feature that reduces the polygon density of proteins as a function of distance to the user, thereby decreasing the computational burden of distant

proteins without affecting perceived detail. It also allows more proteins to be displayed on screen simultaneously.

Protein dynamics can range from localised movement in specific residues to large rearrangements in domains and multiple subunits. Representing these broad spatio-temporal scales is a major challenge for biomedical animators. Protein bond vibrations and domain motions can range from femtoseconds to milliseconds respectively, and in turn, many cellular processes occur in the order of seconds to minutes (*McGill, 2008*; *Miao et al., 2019*). Multi-scale representations in landscape animations of cells often have constrained computer graphics performances, sacrificing atomic resolution and motion.

Cellular environments are heterogeneous, highly dynamic and densely packed. Up to 20% to 30% of intracellular environments is occupied by macromolecular components (*Goodsell et al., 2020*). Molecular crowding is known to influence the association and diffusion of proteins, as well as the rates of enzyme-catalysed reactions (*Mourão et al., 2014*). Yet many visualisations eschew depicting stochastic motion and extreme crowding due to fear of losing focus on the visual narrative or cognitive overload (*Goodsell and Johnson, 2007*; *Jenkinson and McGill, 2012*). Furthermore, the computational expense involved with animating and rendering large numbers of meshes is a significant hurdle. However, underrepresentation or oversimplification has been shown to worsen deep rooted misconceptions, particularly amongst students (*Garvin-Doxas and Klymkowsky, 2008*; *Gauthier et al., 2019*; *Jenkinson et al., 2016*; *Zhou et al., 2008*). Indeed, our inherent ability to picture such

**Video 1.** Animated models of five cellular processes – macropinocytosis (upper right), caveolae (middle left), clathrin coated pits (middle right), exosomes (lower left) and filopodia (bottom lower) – created in Maya. The human figure shown in some of the models is 40 nm tall. *Figure 1* and *Figure 5* provide more information about these processes.

https://elifesciences.org/articles/64047#video1

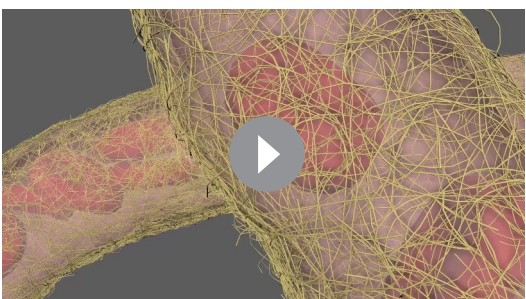

**Video 2.** Animation showing red blood cells flowing through a leaky blood vessel surrounded by basement membrane mesh in the Nanoscape tumour microenvironment (see also *Figure 7*).

https://elifesciences.org/articles/64047#video2

numbers in real life is a challenge, and therefore it is useful to perform 'sanity checks' (*Zoppè, 2017*).

We assessed the feasibility of replicating receptor densities based on empirical data, using the 3D computer graphics application Blender (https://www.blender.org/) and the plugin autoPACK (*Johnson et al., 2015*). The autoPACK algorithm can fill compartmental volumes or surfaces with user defined meshes or protein meshes retrieved from the PDB and has been previously used to generate models of HIV, blood plasma and synaptic vesicles (*Johnson et al., 2015*; *Takamori et al., 2006*).

Receptor density values (number per $\mu m^2$) of six well-known surface biomarkers on MDA-MB-231 cells from flow cytometry data were modelled on 1 $\mu m^2$ area test patches (*Cahall et al., 2015*). The packing simulations revealed 12,650 proteins of variable sizes were easily accommodated with moderate molecular crowding (*Figure 4*). The diversity of surface proteins varies significantly between different cell types, and whilst a fairly equal distribution was modelled for this scenario, proteins are often clustered in functional units or spread unevenly. In addition, the limitations of experimental techniques ought to be scrutinised, for instance, variations in the specificity of antibodies measured with flow cytometry could potentially lead to under- or over exaggeration of numbers. Often, the receptor density data measured is only a snapshot in time, while the receptor population on a cell is highly dynamic and stochastic. Nevertheless, further exploration of experimentally derived population densities using packing algorithms such as autoPACK will improve our understanding of surface protein distribution and organisation.

A negative consequence of choreographed molecular visualisations is that they can often allude to 'directed intent', the most common example being a ligand moving undisturbed towards a receptor, which subsequently leads to immediate binding. A more authentic depiction might be a random walk (constrained by kinetics and thermodynamics) through a packed environment, where the ligand undergoes many nonspecific interactions before binding successfully (*Jenkinson and McGill, 2012*; *Robic, 2010*). Biomedical artists may be disinclined to animate unsuccessful binding events to save on animation and render time, but this will only deepen superficial understanding or promote misconceptions. Whether visual clutter in crowded molecular environments has a negative impact on audiences is a point of contention. Careful use of graphic devices such as titles or arrows, narration and colour can improve complex visualisations and understanding (*Jenkinson et al., 2016*; *Jenkinson and McGill, 2012*; *McClean et al., 2005*).

## Plasma membrane lipids

All eukaryotic cells are surrounded by a plasma membrane consisting of a ~ 4 nm thick lipid bilayer that acts as a semi-permeable barrier between the cell and its environment. It is also a landscape where many signalling events and biological processes take place. Inside cells, specialised compartments are enclosed by lipid membranes to form discrete organelles, segregating their contents from the cytoplasm (*Kobayashi and Menon, 2018*; *van Meer et al., 2008*).

Lipidomics studies have brought great insight into the structure, dynamics and interactions of a variety of lipids within cells (*Ingólfsson et al., 2014*). Lipid bilayers are densely packed with approximately $5 \times 10^6$ molecules per 1 $\mu m^2$ area and can encompass hundreds of different species (*Alberts et al., 2002*). The most common structural lipids in eukaryotic membranes are glycerolipids (~65%), sterols (~25%) and sphingolipids (~10%) (*Shahane et al., 2019*). Whilst cholesterol is generally evenly distributed between the bilayer leaflets, an asymmetric distribution of lipids contributes to their functionality (*van Meer et al., 2008*). The inner leaflet has a greater proportion of phosphatidylethanolamine, phosphatidylserine, and phosphatidylinositol lipids; in contrast, the outer leaflet is more abundant in phosphatidylcholine and sphingolipids (*Kobayashi and Menon, 2018*).

In vivo imaging of membranes is experimentally difficult due to their inherent flexibility and highly dynamic fluctuations. Consequently, computational simulations mainly based on in vitro data have been important for understanding the heterogeneity and dynamics of plasma membranes that may exist in distinct phases, and the organisation of protein complexes therein. Near-atomic models of cell membranes composed of numerous lipid constitutions have been created in silico (*Ingólfsson et al., 2014*). *Figure 3C* depicts a typical lipid bilayer of a cancer cell, with a representative complement of lipid species visualised using the CHARMM-GUI Membrane Builder to highlight some of the differences between the inner and outer leaflets (*Shahane et al., 2019*; *Jo et al., 2008*).

Lipids can move rotationally and laterally within their leaflet and transversely between

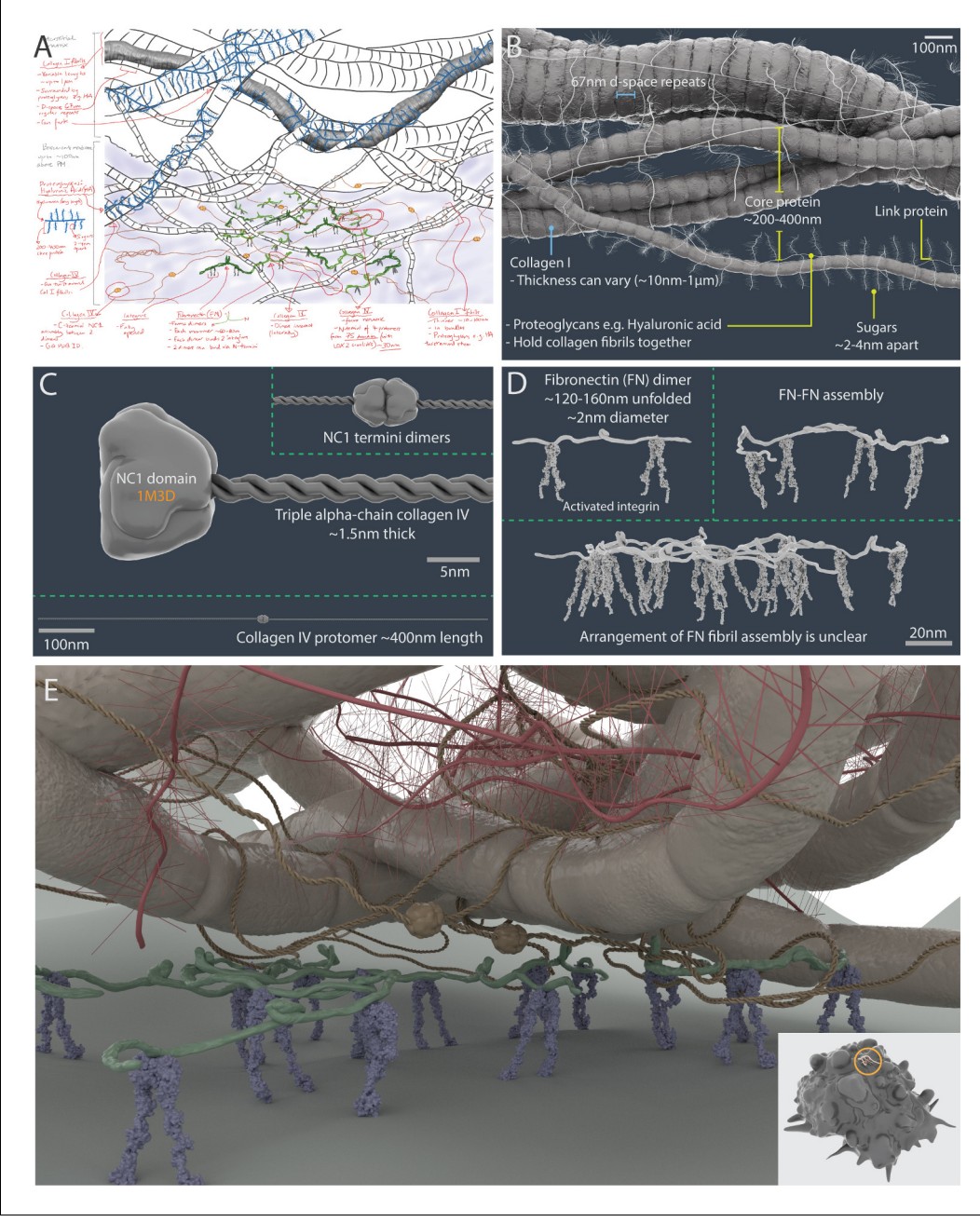

**Figure 6.** A 3D artistic impression of the extracellular matrix in a tumour microenvironment. A pre-production sketch (**A**) and a 3D model (**B**) of collagen I fibrillar bundles and proteoglycans (such as hyaluronic acid). (**C**) Collagen IV protomers and dimers. (**D**) Fibronectin dimers bound to active αVβ3 integrin. (**E**) Artistic interpretation of the extracellular matrix in a tumour microenvironment. The insert shows the scale of the modelled area (circle) relative to a breast cancer cell model (which has a diameter of ~10 μm).

bilayers. Lateral mobility can be expressed by an experimentally determined diffusion coefficient (D). Many lipids have a D value of ~1 μm$^2$ s$^{-1}$, which corresponds to a lipid diffusing a distance of 2 μm within 1 s. In contrast, transverse diffusion or 'flip-flop' is a far slower process that can take hours and is regulated by flippases and floppases to maintain bilayer asymmetry (*Berg et al., 2002*).

Replicating lateral diffusion of millions of lipids at speed in whole-cell and environmental molecular visualisations is not only computationally intensive – it is also questionable whether the viewer can fully observe or appreciate such

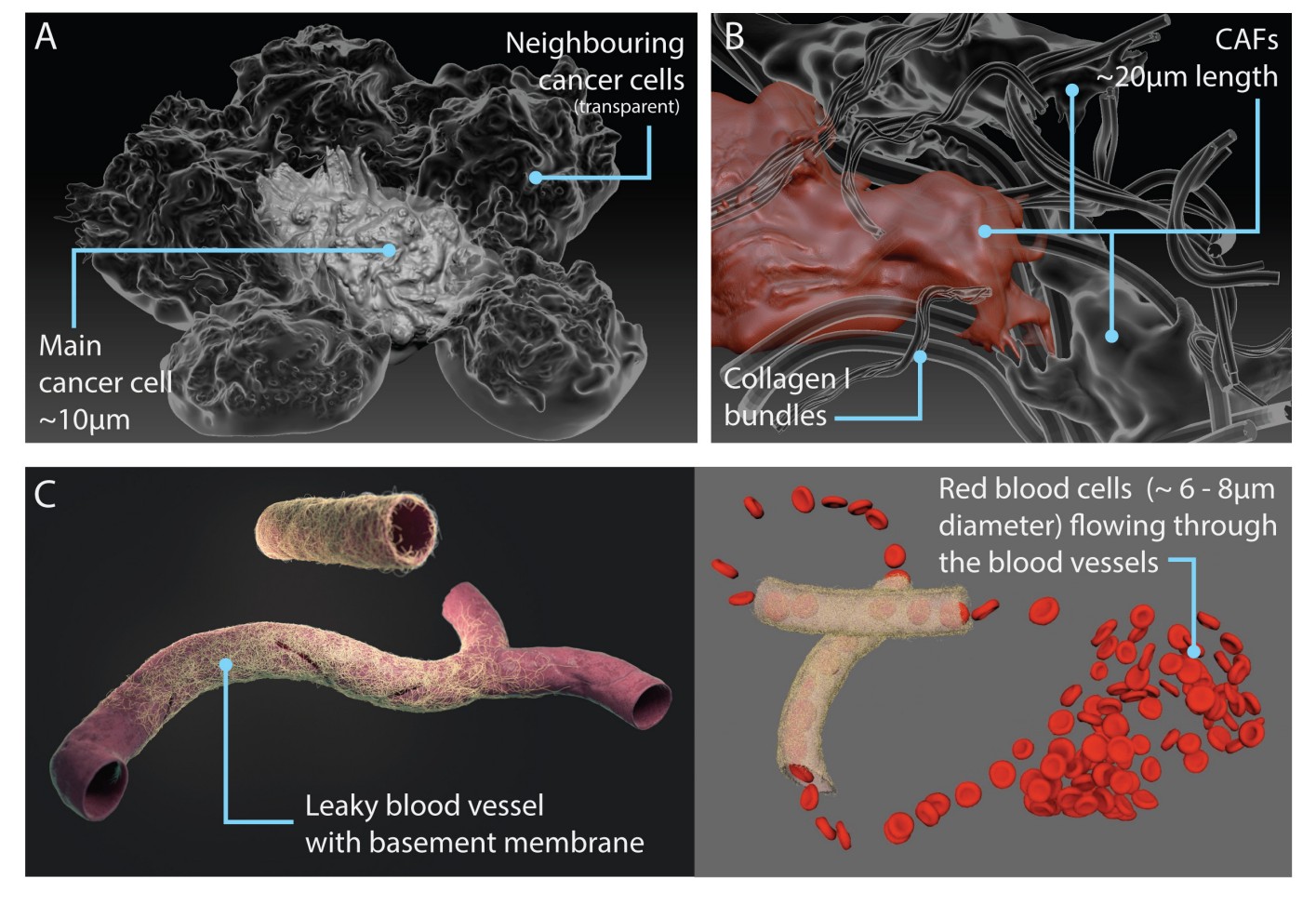

**Figure 7.** Models of components in the tumour microenvironment. (**A**) Additional neighbouring cancer cells (transparent) surrounding the central or main cancer cell. (**B**) Cancer-associated fibroblasts (CAFs) entangled in collagen fibres. (**C**) Leaky blood vessel surrounded by basement membrane mesh (left); snapshot of animation with red blood cells flowing through the vessel (right; see also *Video 2*).

minutiae in a large scene or glean any real insight. To navigate these extensive challenges, the plasma membrane in Nanoscape is instead driven by an animated texture to simulate their form and dynamics.

### Cellular processes

Cellular processes fall within the biological mesoscale, which is an intermediate scale range between nanometre-sized molecular structures and micrometre-sized cellular architecture (*Goodsell et al., 2018*; *Laughlin et al., 2000*). An integrative modelling approach was adopted whereby information from multiple sources was combined for depiction of mesoscopic processes on the cell surface. Surface processes of interest were categorised as either protrusion (filopodia), invaginations (caveolae, clathrin-coated pits, or macropinocytosis) or extracellular vesicles

(exosomes; see *Table 1*). Scanning electron microscopy and transmission electron microscopy data were used to calculate the approximate dimensions of typical breast cancer cells, along with sizes and densities of protrusions and invaginations (*Figure 5*). These measurements were in accordance with published data. Similarly, information on temporal dynamics was taken from research literature (*Table 1*). Since these data fell within a broad temporal range varying from seconds to minutes, cellular features were modelled and animated using 3D software Zbrush and Maya (*Figure 5* and *Video 1*).

### Extracellular matrix

The extracellular matrix (ECM) surrounding cells is an extensive and complex network of structural fibres, adhesion proteins and

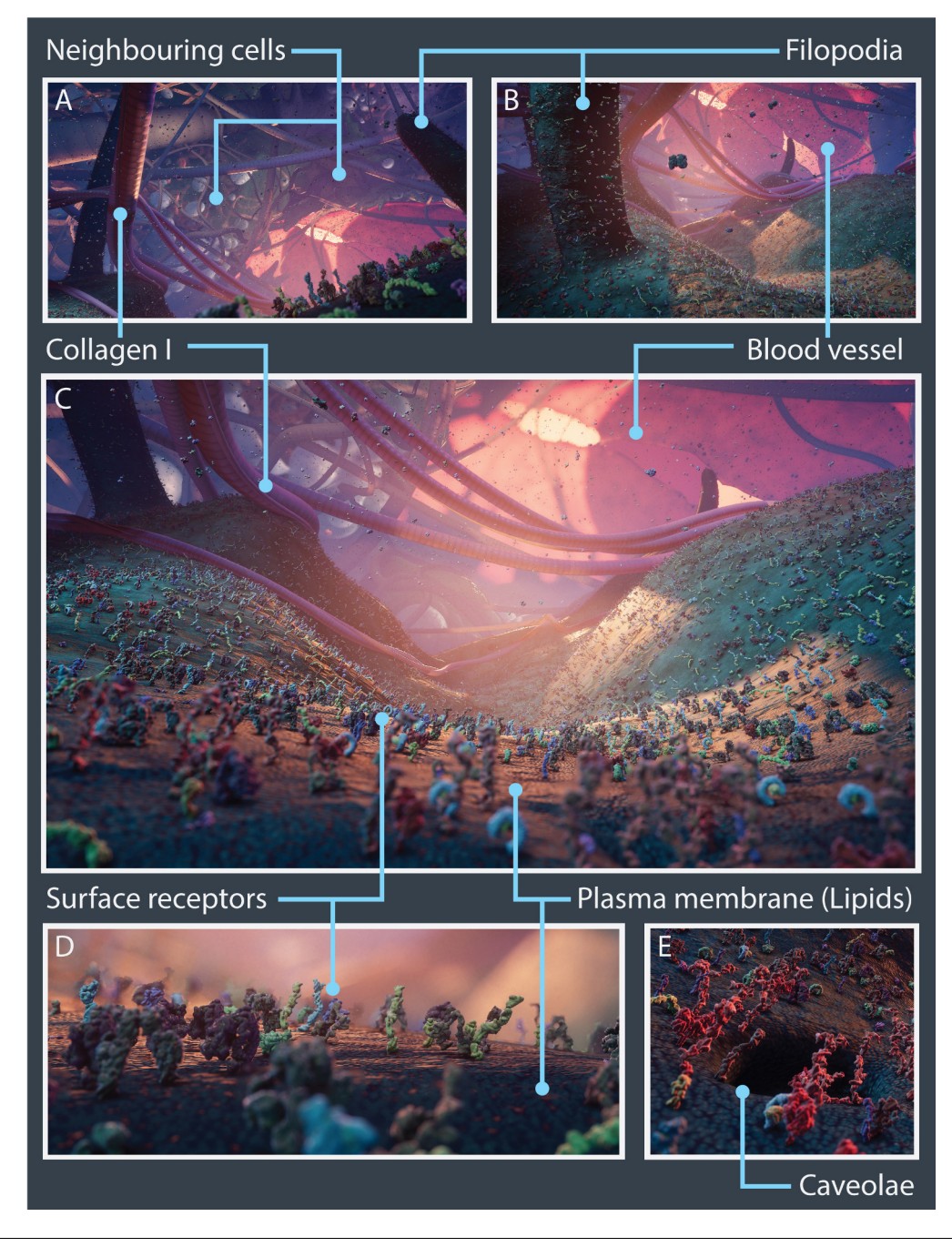

**Figure 8.** Nanoscape real-time open-world experience. Vistas from the Nanoscape real-time open-world experience with key cellular features and microenvironment components highlighted. (A–C) Panoramic views showing surface receptors on the plasma membrane along with neighbouring cancer cells, filopodia, collagen I fibres, and blood vessels. Close-up views of surface receptors and lipids (D) and a caveolae (E).

glycosaminoglycans. It works as a scaffold for the cellular environment and can also influence behaviours, processes and communication of a cell (*Frantz et al., 2010*; *Insua-Rodríguez and Oskarsson, 2016*; *Mouw et al., 2014*). The ECM consists of two biochemically and morphologically distinct forms: the basement membrane, which is a thin layer that forms between epithelial and stromal cells, and the adjacent interstitial matrix, which is a more loosely organised network surrounding cells (*Mouw et al., 2014*).

Despite its vital role, the ECM is often overlooked in many cellular and molecular visualisations due to several reasons. Firstly, high resolution imaging of the ECM in its native state is inherently difficult in opaque tissue. Whilst optical tissue clearing and decellularisation methods are routinely used to enhance the visibility in stained tissues and organs, they are often harsh treatments that may modify the physical and chemical properties of the ECM (*Acuna et al., 2018*). In addition, many in vitro models consisting of a limited mix of collagens and matrix proteins, such as cells suspended in 3D gels, may not always be physiologically relevant.

The components within the matrix are often very large macromolecular complexes, and although complete protein sequences are available on Uniprot, many PDB structures are only of truncated or incomplete proteins (*UniProt Consortium, 2018*). Consequently, it is an arduous task to visualise these large ECM structures in their entirety, and some artistic license may be needed.

To visualise the ECM within a 3D breast tumour microenvironment, we collated information from the literature, including images, and sought advice from experts in the field during the conceptualisation stage (*Figure 6A*; *Table 1*). Although there are a multitude of ECM components that make up the tumour microenvironment, for simplicity it was decided the focus would be only on four key ECM players: collagens I and IV, hyaluronic acid and fibronectin (*Mouw et al., 2014*). In breast cancer, there is usually increased deposition of collagen and fibronectin, and a significant disruption of collagen IV basement membrane networks (*Frantz et al., 2010*).

A stylised artistic approach was adopted to build 3D meshes of the ECM using the modelling program Zbrush. This significantly reduced the polycount of large complex atomic macromolecules, which would otherwise be computationally expensive to render (*Figure 6B–D*; Materials and methods). Following advice from an ECM expert (S. Kadir personal communication with Dr Thomas Cox, Garvan Institute, July 2019), an artistic impression of a tumour microenvironment niche was built, highlighting interactions between integrin molecules on the cell surface and meshes of the basement membrane and interstitial matrix (*Figure 6E*).

Due to significant gaps in knowledge, many visualisation challenges were identified early on during the conceptualisation stage. Exactly how ECM proteins interact with one another to eventually form higher order structures such as fibrils, fibres and ultimately matrices, is very unclear. ECM remodelling is a highly dynamic ongoing process, which involves both proteolytic breakdown of existing matrix components by matrix metalloproteases (MMPs) and deposition of new components by cells. This intrinsic activity has significant implications on tissue development, cell migration and pathologies including cancer (*Frantz et al., 2010*; *Insua-Rodríguez and Oskarsson, 2016*; *Mouw et al., 2014*). However, there is not enough discernible data in the literature to reliably inform anyone trying to represent these processes, and as such, much of this detail was omitted from the ECM represented in Nanoscape.

Many molecular visualisations fail to show connections between the cell surface and the ECM (via cell adhesion molecules binding to ECM components) or veer away from even attempting to represent ECM density in vivo. This may be a combination of incomplete information about precise binding interactions, and a reluctance to complicate a scene for fear of cognitive overload. However, its frequent omission or aggressive simplification will only exacerbate a naive view of the ECM in vivo, whereas there is evidence that more complex molecular representations can in fact improve understanding (*Jenkinson et al., 2016*; *Jenkinson and McGill, 2012*).

### Tumour microenvironment components

In addition to malignant cells and the ECM, a breast tumour microenvironment is made up of a complex mix of cells (including immune cells, fibroblasts, pericytes and adipocytes), blood vessels, lymphatics and various signalling molecules, and many studies show that cancer cells significantly impact their environment; it has also been shown that interactions with non-transformed cells and the tumour vasculature promote the progression of cancer (*Balkwill et al., 2012*; *Quail and Joyce, 2013*; *Insua-Rodríguez and Oskarsson, 2016*). To build a more comprehensive tumour microenvironment, additional breast cancer cells, cancer-associated fibroblasts and a leaky blood vessel with an animated blood flow were incorporated into the Nanoscape scene (*Figure 7*; *Video 2*; Materials and methods).

## The Nanoscape user experience

Nanoscape is distinct from other published molecular and cellular visualisations, being a data-informed artistic innovation that permits user exploration and reflection of a tumour microenvironment. The Nanoscape scene was compiled in the real-time graphics engine Unity3D (https://unity.com/) and can be currently viewed in engine on a desktop gaming PC (with minimum of 1080 GTX GPU). Here, the user is essentially shrunk down to an equivalent height of 40 nm and is able to walk on the surface of a single cancer cell within a discrete play area surrounded by the ECM, neighbouring cancer cells, cancer-associated fibroblasts and a leaky blood vessel to observe surface proteins and cellular processes moving in real-time (*Figure 8*).

Since the user experience was fundamental to the project, it became clear that true depiction of extreme molecular crowding and the ECM had to be compromised. Extraneous complexity such as the density of soluble molecules present in the extracellular space was significantly diminished, water and ions implicit, lipid meshes were substituted with a texture to mimic their form and dynamics, and the ECM reduced to only collagen I fibres, to enable greater visibility of the landscape.

Similarly, it was computationally demanding and aesthetically unfavourable to replicate the broad temporal ranges of both molecular and cellular processes. Whilst the dynamics of the major cellular processes could be animated accurately, atomic resolution was sacrificed, and protein conformational movement was appreciably slowed down to enable viewers to observe protein-protein interactions more clearly. To partially compensate for these diminutions, some artistic license was applied to evoke the 'feel' of a densely packed cancerous milieu through use of colour, lighting, and sound, even if nanoscale cellular entities are below the wavelength of light and devoid of noise.

Nanoscape has significantly surpassed the level of cell surface detail and complexity found in Journey to the Centre of the Cell. And although it is not fully immersive or interactive as the virtual reality experience of Journey to the Centre of the Cell, it has been carefully choreographed to engage the viewer to appreciate the heterogeneity of a tumour microenvironment, without overwhelming them with minutiae and promote understanding of some cell biology fundamentals. Future work involves the development of user-defined control features,

which empower the viewer to focus in or out of regions to switch between observing atomic detail to large cellular processes, and adjustable dials for regulating molecular density and temporal dynamics, to experience a more authentic cellular environment.

### Evaluation and broader applications

Though outside the scope of the development stage of the project, formal evaluation of the user experience and objective outcomes is necessary and will inform the practicality of this application as an educational aid. In particular, what effect does increasing the authenticity of the visualisations (with respect to biological densities, scales and interactions) have on users' fundamental understanding of cell biology? In future studies it will be prudent to test several versions of the application varying these parameters and assess the students' comprehension of core concepts, such as the effect of receptor density on ligand biding.

It is hypothesised that increasing the authenticity of visualisations will improve the comprehension of these advanced topics, as has been observed in similar studies (*Jenkinson and McGill, 2012*). Both qualitative and quantitative aspects of the user experience, including visual attention (through eye-tracking software), analytics such as time spent in specific locations, and subjective measurements of the users' visual and mental load, can also be assessed while varying these parameters.

Further analysis might also investigate whether learning outcomes can be influenced by the delivery platform of this application, evaluating the interactive PC experience or virtual reality immersive experience against similar content delivered in a more linear narrative, such as standard video format. Allowing users to control what aspects of the landscape they view and interact with may help them to focus on specific gaps in knowledge. This data is likely to inform science educators on the most appropriate level of detail and delivery platform for each learning objective.

Beyond the didactic purpose of the application, Nanoscape is well-poised to be a valuable public outreach tool. In addition to rich scientific detail, thoughtful consideration was given to key design elements such as colour theory and sound design. These arts-led influences allow Nanoscape to sit alone as a piece of science-based art, inviting the user to wonder at the chaotic molecular landscapes within the body. Whether this type of visualisation offers benefits

to the wider scientific research community is yet to be seen. While the artistic and computer graphics-related treatments of proteins and cellular components in Nanoscape is unlikely to make this application suitable for rigorous data interrogation, it may stimulate more holistic reflection of biological systems. Applications that allow multimodal data exploration are promising and may promote broader scientific speculation than each modality examined separately.

### Data collection archiving

Created in 1971, the PDB remains the largest international repository for experimentally determined atomic structures. Considerable progress has been made in recent years to develop platforms and standards for archiving, validating, and disseminating new biological models defined by the PDBx/mmCIF dictionary (*Young et al., 2018*). Integrative or hybrid modelling structures are, however, not currently included in the PDB because data standards for archiving have not yet been implemented. Hence, the PDB-Dev was established in 2018 as a prototype archiving system (https://pdb-dev.wwpdb.org) (*Vallat et al., 2018*). This repository contains embedded 3D viewers, links to download structures and related database entries, but also includes citations, input data and software used in the creation of models. Whilst many biomedical animators take models from such sources, clear citation of their molecular visualisation content is often limited.

The information gathered during the pre-production phase of Nanoscape is summarised in *Figure 1*, *Table 1* and Appendix 1 (which includes the PDB IDs for the proteins). Furthermore, Appendixes 2–9 document the methodology behind the creation of mMaya mechanism-of-action animations for the following receptors: EGFR and Her3 (Appendix 2; *Kovacs et al., 2015*); αVβ3 integrin (Appendix 3; *Zhu et al., 2008*; *Chen et al., 2011*); VEGFR1 (Appendix 4; *Sarabipour et al., 2016*); c-KIT (Appendix 5; *Felix et al., 2015*); insulin receptor (Appendix 6; *Gutmann et al., 2018*); Tetraspanin CD81 (Appendix 7); the TNFR superfamily of receptors (Appendix 8; *Naismith et al., 1996*; *Vanamee and Faustman, 2018*); and GLUT1 (Appendix 9). These appendixes also include associated references, and information on the artistic approaches taken for modelling assets, along with explanatory comments.

Archiving the information taken from a variety of literature, databases and communication with experts in the field and presenting it in an accessible format will enable others to freely scrutinise or validate the work impartially, and to potentially build new, improved future versions. Selecting and curating vast amounts of information is, however, extremely time consuming, requires an aptitude for interpreting scientific data, and is a constant race to keep abreast of the latest discoveries. Similarly, sustaining up-to-date versions of large-scale complex projects such as Nanoscape whenever new data becomes available, or existing information is proved redundant, is a huge challenge and only reinforces the need for transparent systems of citation in scientific visualisations and comprehensive procedural frameworks to seamlessly implement new models or animations.

## Conclusions

Nanoscape is an innovative collaboration that has produced a multi-scale, explorable 3D environment of a cell. Our work sheds light on some of the technical and creative processes, decisions and sacrifices made in depicting cell surface entities and dynamics as close to experimental data as possible, whilst balancing concerns for the user experience and visual aesthetics. Although initially its main purpose was to be a unique educational and outreach tool to communicate some of the complexities of a tumour microenvironment, the final visualisation experience may also help experimentalists to reflect upon their own data.

Integrative modelling and visualisation of biomolecular systems and multi-scale cell models are becoming increasingly sophisticated, and immersive, virtual field trips to a cell environment such as Nanoscape may provide insights into function and behaviour of a cell. As computer hardware and software continues to evolve to cope with processing enormous amounts of data, improved visual or interactive 3D representations may one day lead the way for scientists to perform in silico experiments and potentially help with the development of new drugs.

Ideally, such a model would fully reflect the spatio-temporal complexities and heterogeneity of the entire cell and its environment and would be capable of continuous iterations and be falsifiable. To accomplish such an ambitious feat, researchers and developers from multiple scientific, computer graphics and design fields must work together.

## Materials and methods

### Software

Commercially available 3D computer graphics software for animation and modelling, Maya (https://autodesk.com/maya), the plugin Molecular Maya (mMaya; https://clarafi.com/tools/mmaya/), and Zbrush (http://pixologic.com/features/about-zbrush.php) were used to build and animate assets. The 3D procedural software Houdini (https://www.sidefx.com/products/houdini/) and the cross-platform game engine Unity3D (https://unity.com/) were used to compile the Nanoscape open-world environment; see 'Nanoscape open-world compilation' below for details. Unless stated, all images were rendered in Maya using Arnold at 4K (3840 × 2160).

The Unity3D game engine was used to assemble the various components into a coherent representation of a cell surface environment. Static assets were imported into the engine by standard methods. The animated cellular processes and horde of proteins were integrated and simulated in Houdini and output as custom caches that are streamed into Unity3D at runtime.

### Proteins

Protein structures were retrieved from the RSCB Protein Data Bank (PDB), and mechanism-of-action animations were simulated using the mMaya modelling and rigging kits. Rigged surface or backbone meshes were extracted, and animation playblasts recorded in Maya and composited in After Effects (Adobe CC). See Appendixes 1–9 and 'Surface protein simulations' below for details.

For the creation of stylised proteins, polygonal backbone meshes extracted from PDB structures were sculpted and textured in Zbrush, with textures exported for later use as detail maps in the rendering software. Images were rendered in Maya using Arnold at 1400 × 1400.

Receptor density simulations on 1 μm² surface areas (sphere and plane) based on MDA-MB-231 cells from flow cytometry data were created in Blender 2.78 using the plugin autoPACK (autopack.org) with the spheresBHT packing method (*Cahall et al., 2015*; *Johnson et al., 2015*). Low poly PDB meshes of CD44, EGFR, EpCAM, Her2, ICAM1 and αVβ3 integrin were created in mMaya.

### Lipids

A lipid bilayer consisting of 400 lipids was simulated using the CHARMM-GUI Membrane Builder (*Jo et al., 2008*) based on data from Table 2 in *Shahane et al., 2019*. UCSF Chimera was used to export lipid meshes to Maya (*Pettersen et al., 2004*).

### Cells, cellular processes and ECM models

Information on dimensions and temporal dynamics were taken from the literature and data from our collaborators (See *Table 1*). All 3D assets were modelled in Zbrush or Maya. Cellular processes were animated in Maya.

### Surface protein simulations

Mechanism-of-action (MoA) animations for nine selected receptors were made using the mMaya Modelling and Rigging Kits (https://clarafi.com/tools/mmaya/). mMaya simulations are qualitative and were used to inform the artistic design about general conformational changes and movements that may occur in receptor-ligand binding events. The mMaya Rigging Kit relies on Molecular Mechanics force fields to model molecular structures and their interactions and uses Maya's particle system and nucleus solver to simulate structural changes and interactions. Molecular Mechanics force fields model how atoms can interact through simplified potentials, and provide a set of spatial restraints (distance, angular, dihedral) to preserve the stereochemistry of each chemical building block imported into the software. By using these force field parameters and molecular topologies, the Rigging Kit builds molecular 'rigs' and provides a set of tools for users to apply various external forces interactively. Unlike molecular dynamics simulations, which follow strict thermodynamic laws and make them suited to explore the unsupervised evolution of a molecular system, the mMaya Rigging Kit provides a molecular simulation environment for atomic and coarse-grained molecular models that avoids steric clashes while allowing users to orchestrate complex transitions and large conformational changes.

All rigs were 'all atom no hydrogen' constructs and conformational changes were simulated by either target morphing between two endpoint PDB structures (usually inactive and active states), for example EGFR (1NQL → 3NJP) (see Appendix 2, which includes a MoA animation for EGFR), or manually moving handles added to selected regions of the protein rig (e.g., domains). For some proteins where only one PDB structural state was available, as with Her3 (inactive, 1M6B), movement was approximated by targeting protein domains to morph

into the conformational state of another known family member (in this case active Her4 3U7U; see Appendix 2, which includes a MoA animation for Her3). In addition, simulations were inferred from MoA hypotheses published in the literature, and rig handles were applied to manipulate the movement of protein rigs to create a 'hypothetical' conformation, as in the case of αVβ3 integrin extended form conformation (see Appendix 3, which includes a MoA animation for αVβ3 integrin). When morph targets were set between a rig and a PDB chain, the PDB chain was positioned manually to be near the rig i.e., the C-termini of the chains were aligned as close as possible. Elastic networks were made for protein domains and the strength adjusted, if necessary, to maintain the domain structures. Rig environmental dynamics were adjusted accordingly (e.g., turbulence field magnitude and damping) and simulations tested until the rig moved in a 'smooth' manner. Final rig simulations were cached, and receptor surface or backbone meshes extracted. Ligand binding events were added later by key-framing the motion of the meshes manually. In some of the MoA animations, the ligand binding event was omitted and only the resultant conformational change in the protein was shown.

### ECM asset creation

Collagen I is the most abundant structural component of the interstitial membrane. Protein fibrils of varying thicknesses (10 nm–1 μm) were sculpted to highlight striated 67 nm d-space repeats and arranged into bundles to represent fibres. A bespoke insert mesh zBrush tool was created to wrap the proteoglycan hyaluronic acid around collagen I structures (*Figure 6B,E*).

Similarly, an insert mesh zBrush tool was made for modelling Collagen IV protomers, which consisted of three intertwining α-chains that form a triple helix 400 nm in length. Two collagen IV protomers were joined head-to-head via NC1 dimers (PDB structure 1M3D) at the C-termini, and four collagen N-termini overlaid to form the 7S domain (28 nm overlaps), to create an extensive branched mesh network (*Figure 6C,E*).

Fibronectin monomers are made up of three repeating units (FN types I, II, III) and usually form dimers linked by a pair of disulphide bonds at their c-termini (*Pankov and Yamada, 2002*). Fibronectin dimers are long and can form fibrils (ranging from ~133–190 nm) (*Früh et al., 2015*). However, the precise molecular arrangement and their associations with multiple binding partners is still unclear. Alternative splicing can lead to over 20 protein variants in humans (*Pankov and Yamada, 2002*). Therefore, a simplified dimer mesh was modelled, which was shown only bound to integrin (*Figure 6D,E*).

### Cancer cells, cancer-associated fibroblasts and blood vessel

Tumour microenvironment components: additional neighbouring cancer cells (~10 μm diameter), cancer-associated fibroblasts (~20 μm length) and a leaky blood vessel (~10 μm diameter) with red blood cells (~6–8 μm diameter) were modelled in zBrush based on various microscopy images taken from the literature (See *Figure 7*). The blood flow in the vessel was animated in Maya (*Video 2*).

### Acknowledgements

This work was funded by the Australian Research Council (ARC) Centre of Excellence in Convergent Bio-Nano Science and Technology (CBNS) and a National Health and Medical Research Council of Australia fellowship to RGP (APP1156489). The authors thank CBNS Director Professor Tom Davis for supporting the project, and CNBS collaborators Dr Angus Johnston (Monash University) and Professor Kristofer Thurecht (University of Queensland) for their advice. We are grateful to Dr Thomas Cox (Garvan Institute) for his expertise on the ECM, and to Nick Maurer for his help optimising performance out of the Unity3D game engine at the last minute. A special thanks to Mark Arrebola for advice and additional support during the project. The authors acknowledge the use of the Microscopy Australia Research Facility at the Center for Microscopy and Microanalysis (University of Queensland) and the assistance of Mr Rick Webb.

**Shereen R Kadir** is in the 3D Visualisation Aesthetics Lab, School of Art and Design, and the ARC Centre of Excellence in Convergent Bio-Nano Science and Technology, University of New South Wales, Sydney, Australia

https://orcid.org/0000-0002-3960-988X

**Andrew Lilja** is in the 3D Visualisation Aesthetics Lab, School of Art and Design, and the ARC Centre of Excellence in Convergent Bio-Nano Science and Technology, University of New South Wales, Sydney, Australia

https://orcid.org/0000-0001-8311-3702

**Nick Gunn** is in the 3D Visualisation Aesthetics Lab, School of Art and Design, and the ARC Centre of Excellence in Convergent Bio-Nano Science and

Technology, University of New South Wales, Sydney, Australia

**Campbell Strong** is in the 3D Visualisation Aesthetics Lab, School of Art and Design, and the ARC Centre of Excellence in Convergent Bio-Nano Science and Technology, University of New South Wales, Sydney, Australia

**Rowan T Hughes** is in the 3D Visualisation Aesthetics Lab, School of Art and Design, and the ARC Centre of Excellence in Convergent Bio-Nano Science and Technology, University of New South Wales, Sydney, Australia

https://orcid.org/0000-0001-5618-381X

**Benjamin J Bailey** is in the 3D Visualisation Aesthetics Lab, School of Art and Design, and the ARC Centre of Excellence in Convergent Bio-Nano Science and Technology, University of New South Wales, Sydney, Australia

**James Rae** is in the Institute for Molecular Bioscience, ARC Centre of Excellence in Convergent Bio-Nano Science and Technology, and Centre for Microscopy and Microanalysis, University of Queensland, Brisbane, Australia

**Robert G Parton** is in the Institute for Molecular Bioscience, ARC Centre of Excellence in Convergent Bio-Nano Science and Technology, and Centre for Microscopy and Microanalysis, University of Queensland, Brisbane, Australia

https://orcid.org/0000-0002-7494-5248

**John McGhee** is in the 3D Visualisation Aesthetics Lab, School of Art and Design, and the ARC Centre of Excellence in Convergent Bio-Nano Science and Technology, University of New South Wales, Sydney, Australia

john.mcghee@unsw.edu.au

https://orcid.org/0000-0002-9264-7535

*Author contributions:* Shereen R Kadir, Conceptualization, Data curation, Formal analysis, Investigation, Visualization, Methodology, Writing - original draft, Project administration, Writing - review and editing; Andrew Lilja, Conceptualization, Data curation, Investigation, Visualization, Methodology; Nick Gunn, Campbell Strong, Rowan T Hughes, Conceptualization, Software, Methodology; Benjamin J Bailey, Conceptualization; James Rae, Visualization; Robert G Parton, Conceptualization, Funding acquisition, Writing - review and editing; John McGhee, Conceptualization, Supervision, Funding acquisition, Project administration, Writing - review and editing

*Competing interests:* The authors declare that no competing interests exist.

## Funding

| Funder | Grant reference number | Author |
|---|---|---|
| ARC Centre of Excellence in Convergent Bio-nano Science & Technology | CE140100036 | Shereen R Kadir Andrew Lilja Nick Gunn Campbell Strong Rowan T Hughes Benjamin J Bailey James Rae Robert G Parton |

The funders had no role in study design, data collection and interpretation, or the decision to submit the work for publication.

**Decision letter and Author response**
Decision letter https://doi.org/10.7554/eLife.64047.sa1
Author response https://doi.org/10.7554/eLife.64047.sa2

## Additional files
### Supplementary files
• Transparent reporting form

## Data availability
All data generated or analysed during this study are included in the manuscript and supporting files. Protein structures were sourced from the publicly accessible Protein Data Bank.

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

# Appendix 1

## Surface protein structures associated with breast cancer cells

| Receptor/soluble protein | Ligand | PDB Structures | Description |
|---|---|---|---|
| EGFR (HER1, ERBB1)[2] | EGF or TGF-α | 1NQL | EGFR monomer (unliganded) |
| | | 3NJP | EGFR dimer (EGF bound) |
| | | 1MOX | EGFR (TGF-α bound) |
| HER2 (ERBB2) | No known ligand | 1S78 | Her2 Monomer (unliganded) |
| HER3 (ERBB3)[2] | Neuregulins (NRG1 + NRG2) | 1M6B | Her3 Monomer (unliganded) |
| HER4 (ERBB4) | NRG1 | 2AHX | Her4 Monomer (unliganded) |
| | | 3U7U | Her4 Monomer (NRG1 bound) |
| Integrin αVb3[3] | Various e.g. ECM proteins | 4G1M | alpha V beta three structure (inactive) |
| VEGFR-1[4] | VEGF-A | VEGFR-1_D1-7_VEGF-A_composite_model | Full-length structure of the VEGFR-1 ectodomain (VEGF bound) |
| c-KIT[5] | SCF | 2EC8 | c-KIT monomer (unliganded) |
| | | 2E9W | c-KIT homodimer (SCF bound) |
| Insulin Receptor (IR)[6] | Insulin | 4ZXB | Insulin Receptor heterodimer (unliganded) |
| | | 6CEB | Insulin Receptor heterodimer ectodomain (two insulin molecules bound) |
| Insulin-like growth factor receptor (IGF1R) | IGF-I | 5U8Q | Type one insulin-like growth factor receptor heterodimer (IGF-I bound) |
| MHC I (aka HLA-B27) | N/A (presents antigens) | 1HSA | HLA-B27 and beta-2 microglobulin (model peptide sequence bound) |
| TLR4-MD2 complex | LPS (lipopolysaccharide) | 3FXI | Toll-like receptor four and MD-2 (lipopolysaccharide of Gram-negative bacteria bound) |
| TLR1-TLR2 heterodimer | Lipopeptide | 2Z7X | TLR1-TLR2 heterodimer (tri-acylated lipopeptide bound) |
| CXCR4 | CXCL12 (SDF-1) | 3OE0 | CXCR4 chemokine receptor in complex with small molecule antagonist IT1t |
| | | 2K05 | SDF1 in complex with the CXCR4 N-terminus |
| Tetraspanin CD81[7] (TAPA-1) | Cholesterol | 5TCX | Tetraspanin CD81 |
| TNFR1[8] | TNF, LTα, LTß | 4MSV | FASL and DcR3 complex (used instead as TNFR1 is incomplete) |
| | | 2NA7 | Transmembrane domain of human Fas/CD95 death receptor |
| | | Stem Linker | Constructed in mMaya from sequence |
| LDL receptor-PCSK9 complex | lipoproteins | 1N7D | Extracellular domain of the LDL receptor at endosomal pH (no lipoprotein bound) |
| | | 3P5C | PCSK9-LDLR structure at neutral pH (no lipoprotein bound) |
| CSF1R | CSF1 | 4WRL | Dimeric CSF-1 bound to two CSF-1R molecules (D1–D3 part only) |
| | | 4WRM | CSF-1R monomer (CSF-1 bound) |

*Continued on next page*

*continued*

| Receptor/soluble protein | Ligand | PDB Structures | Description |
|---|---|---|---|
| Transferrin Receptor (TfR) | Transferrin (Tf) | 1SUV | Transferrin Receptor (Serotransferrin, N-lobe and C-lobes bound) |
| EphA2 | EphrinA5 | 2X10 | EphA2 ectodomain (monomer) |
| | | 3MX0 | EphA2 ectodomain (in complex with ephrin A5) |
| EphA4 | EphrinA5 (and others) | 4M4P | EphA4 ectodomain (monomer) |
| | | 4M4R | Epha4 ectodomain (complex with ephrin A5) |
| MMP-2 (Gelatinase A) | Various e.g. ECM proteins | 1CK7 | MMP-2 (Gelatinase A) |
| MMP-9 (Gelatinase B) | Various e.g. ECM proteins | 1L6J | MMP-9 (Gelatinase B) |
| TIMP-2 (tissue inhibitors of metalloproteinases-2) | MMPs | 1GXD | proMMP-2/TIMP-2 complex |
| CD44 | glycosaminoglycan hyaluronan, collagens, osteopontin, MMPs | 2JCP | Hyaluronan binding domain of murine CD44 |
| | | Standard stem region | Constructed in mMaya from sequence |
| EpCAM | EpCAM (on other cells) | 4MZV | EpCAM cis-dimer |
| N-Cadherin | N-Cadherin (on other cells) | 3Q2W | N-Cadherin trans dimer |
| E-Cadherin | E-Cadherin (on other cells) | 3Q2V | E-Cadherin trans dimer |
| ICAM | LFA-1 (an integrin present on leukocytes) | 1IC1 | ICAM-1 D1-D2 domains |
| | | 1P53 | ICAM-1 D3-D5 domains |
| Glucose transporter GLUT-1[9] | Glucose | 4PYP | GLUT1 (inward-open; glucose-bound) |
| | | 4ZWC | GLUT3 (outward-open; maltose-bound) |
| Potassium Channel Kv1.2 | K+ | 3LUT | Full-length Shaker Potassium Channel Kv1.2 |
| Na+/K + pump | Na+, K+ | 2ZXE | Sodium-Potassium pump |
| ABCA1 | phospholipids and cholesterol | 5XJY | Lipid Exporter ABCA1 |

Column one lists the protein; column two lists the ligand; column three lists the PDB ID for the protein in the RSCB protein data bank (PDB); column four contains a general description of the content of the PDB file. A superscript in column one indicates that more information, including a movie, is available in one of the appendixes (with a superscript of 2 indicating that this information is available in Appendix 2, and so on). More information is also available in *Figure 2*.

## Appendix 2

## Mechanism-of-action animations for the ErbB family of surface receptors

The ErbB family of proteins (*Kovacs et al., 2015*) contains four receptor tyrosine kinases, including epidermal growth factor receptor (EGFR; also known as ErbB1, or Her1 in humans). The other three members of the family are ErbB2, ErbB3 and ErbB4 (which are known as Her2, Her3 and Her4 in humans; see *Figure 3*).

All members of the ErbB/Her family have four extracellular domains (I–IV).

- *Appendix 2—figure 1A* shows inactive EGFR (left) and active EGFR (right).
- *Appendix 2—figure 1B* shows inactive Her3 (left) and active Her4 (right).
- In the monomeric tethered conformation (EGFR, Her3 and Her4), the dimerization arm (in domain II) is completely occluded by intramolecular interactions with domain IV.
- Ligand binding causes a conformational change ~ 130° rotation of domains I + II with respect to domains III + IV (*Appendix 2—figure 1*), and leads to dimerization (homo- and hetero-dimerization with Her family members).
- The dimerization arm of one monomer interacts with the corresponding element of the dimer partner.
- Her2 exists in the extended conformation without ligand binding

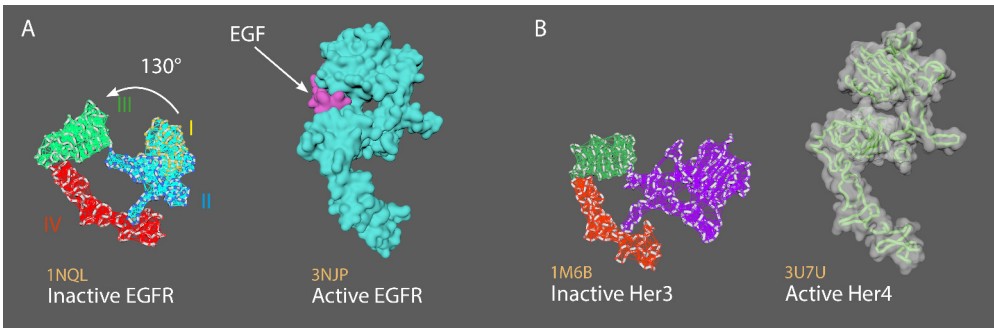

**Appendix 2—figure 1.** Two members of the ErbB family of surface receptors. (**A**) Inactive (left) and active (right) EGFR. (**B**) Inactive Her3 (left) and active Her4 (right).

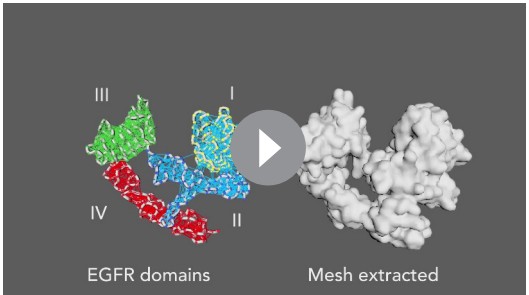

**Appendix 2—video 1.** Animation strategy for EGFR.          1NQL = inactive EGFR (monomer; *Ferguson et al., 2003*). 3NJP = active EGFR (dimer; *Lu et al., 2010*). Rigged 1NQL with mMaya rigging kit, created elastic networks for each domain: I + II combined, III, and IV. Set 3NJP as a target for 1NQL to morph into, and cached the animation. See 'Surface protein simulations' in Materials and methods for more details.

https://elifesciences.org/articles/64047#A2video1

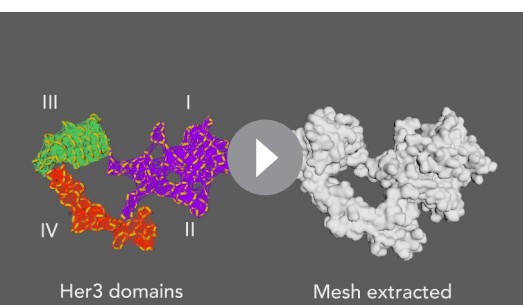

**Appendix 2—video 2.** Animation strategy for Her3. 1M6B = inactive Her3 (monomer; *Cho, 2002*). 3U7U = active Her4 (dimer; *Liu et al., 2012*). Rigged 1M6B with mMaya rigging kit, created elastic networks for each domain: I + II combined, III, and IV. Set 3U7U as a target for 1M6B to morph into, and cached the animation. See 'Surface protein simulations' in Materials and methods for more details.

https://elifesciences.org/articles/64047#A2video2

## Appendix 3

### Mechanism-of-action animations for the surface receptor αVβ3 integrin

- *Appendix 3—figure 1* shows the three major conformational states of integrin in the switch-blade model: (A) bent with closed headpiece; (B) extended with a closed headpiece; (C) extended with an open headpiece (*Zhu et al., 2008*; *Chen et al., 2011*).
- *Appendix 3—figure 1* also shows the four extracellular domains in alpha chains (αV; lower left) and eight in beta chains (β3; lower right).
- Ligands vary (e.g. ECM proteins).

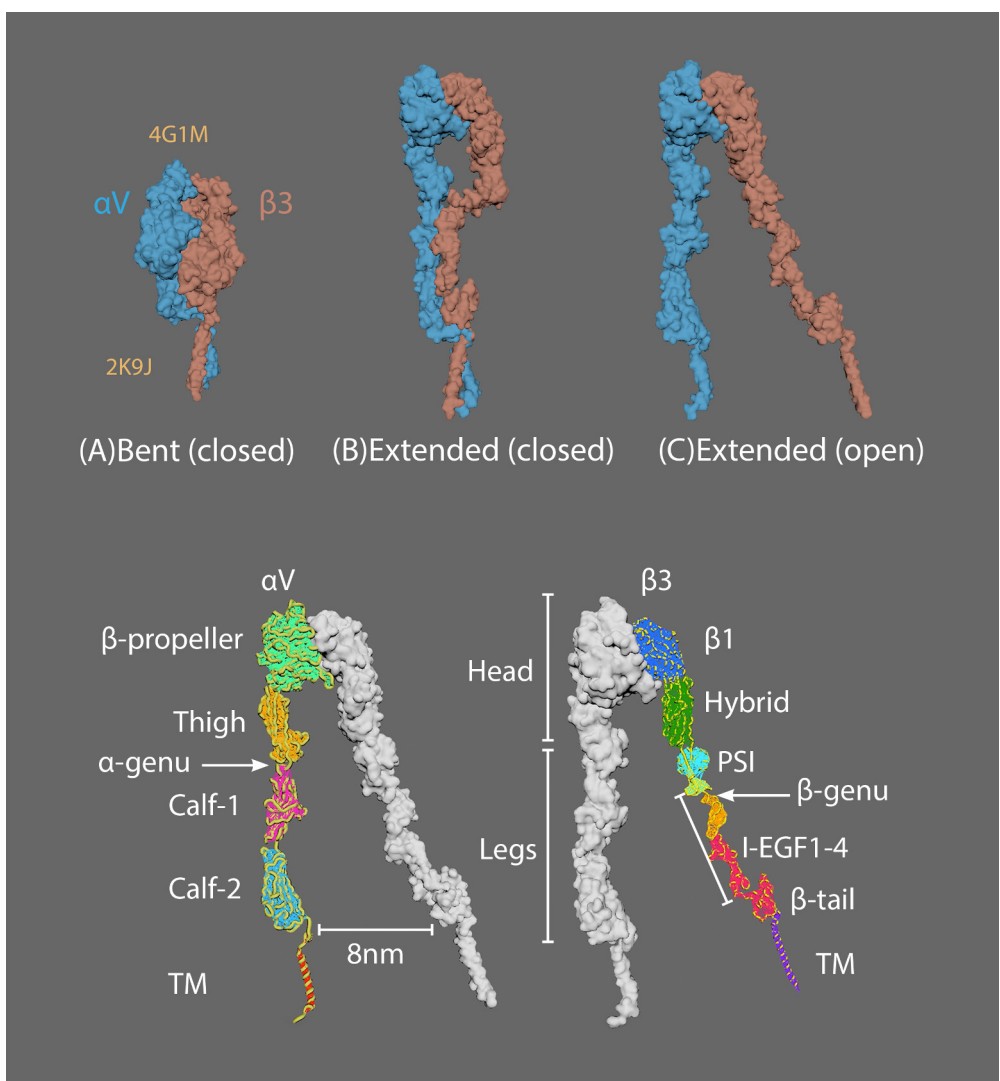

**Appendix 3—figure 1.** The three major conformational states of integrin. (**A**) Bent with closed headpiece; (**B**) extended with a closed headpiece; (**C**) extended with an open headpiece. Extracellular domains in alpha chains (lower left) and beta chains (lower right).

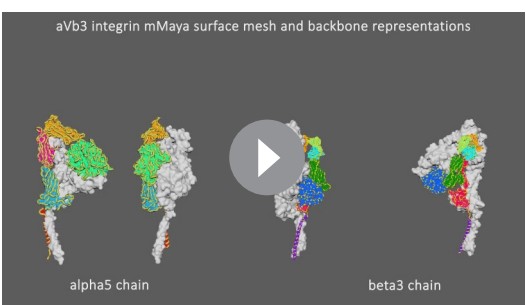

**Appendix 3—video 1.** Animation strategy for αVβ3 integrin. 4G1M = αVβ3 inactive bent conformation (*Dong et al., 2012*). 2K9J = transmembrane domain of integrin αIIb-β3 (*Lau et al., 2009*). Connected 2K9J to the C-term of αVβ3 integrin (4G1M) using the mMaya modelling kit. New PDB created was rigged and elastic networks made for all extracellular domains. As no extended structures are available artistic licence was used to create hypothetical structures. Handles were created for the head portion of αV and β3 to manually open the mesh into the extended closed conformation, and a second handle created for the I-EGFR1-4 and β-tail to move ~8 nm into the extended open conformation.

https://elifesciences.org/articles/64047#A3video1

## Appendix 4

## Mechanism-of-action animations for the surface receptor VEGFR1

- VEGFR1 has seven Ig-like domains (D1-D7) and binds VEGF-A (dimer; *Sarabipour et al., 2016*).
- *Appendix 4—figure 1* shows two views of the full composite dimer, which is built from 5T89 (VEGFR-1 Domains 1-6) and 3KVQ (VEGFR-2 Domain 7; *Markovic-Mueller et al., 2017*; *Yang et al., 2010*).
- D2 and D3 ectodomains are binding sites for VEGF-A.
- D3/VEGF-A binding triggers interactions between D4–5 and D7 in VEGFR homodimers.

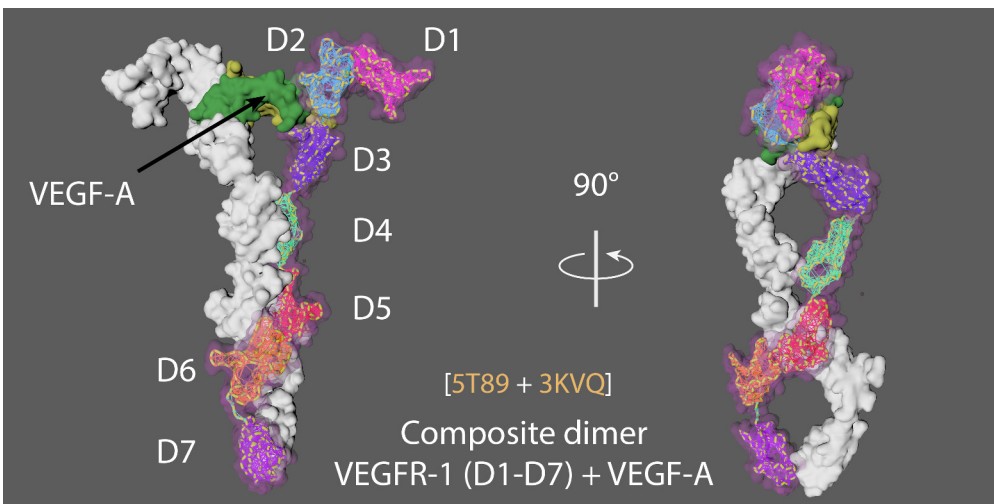

**Appendix 4—figure 1.** Two views of the VEGFR1–VEGF-A composite dimer.

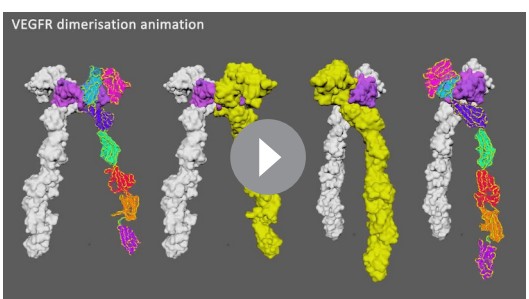

**Appendix 4—video 1.** Animation strategy for VEGFR1.    The composite VEGFR1 dimer was built from 5T89 and 3KVQ (see *Appendix 4—figure 1*) and rigged using mMaya rigging kit. Elastic networks were created for each domain (D1 to D7). Handles were made for the entire CA backbone and individual domains to manually pull the rig away from the dimer conformation, artistic licence was used to create a hypothetical monomer conformation (as no monomer structure is currently available). The hypothetical monomer was simulated to conformationally morph into the target composite VEGFR dimer for the cached animation.

https://elifesciences.org/articles/64047#A4video1

## Appendix 5

### Mechanism-of-action animations for the surface receptor c-KIT

- Stem Cell Factor (SCF) protomer binds directly to the D1, D2, and D3 ectodomain of c-KIT, which leads to the dimerization of two c-KIT molecules (*Appendix 5—figure 1*; *Felix et al., 2015*).
- This results in the lateral interaction of Ig-like domains D4 and D5 (i.e. D4 to D4 and D5 to D5) between the two monomers and brings the c-termini of the monomers 15 Å of each other.

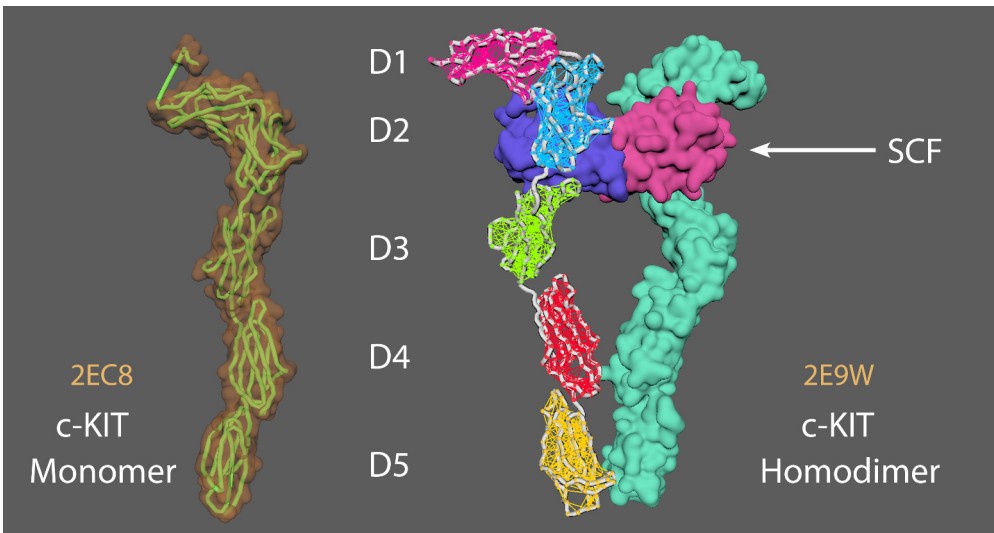

**Appendix 5—figure 1.** A c-KIT monomer (left) and a c-KIT homodimer (right) after binding by a Stem Cell Factor (SCF) protomer.

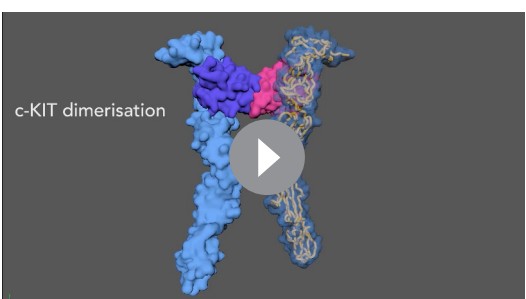

**Appendix 5—video 1.** Animation strategy for c-KIT.     2EC8 = c KIT monomer (*Yuzawa et al., 2007*). 2E9W = c KIT homodimer with SCF (*Yuzawa et al., 2007*). Aligned chain A from 2E9W (dimer conformation) to 2EC8 (monomer conformation). Saved a new PDB version of chain A 2E9W (now as a monomer conformation). Created elastic networks for each domain (D1-5). Set it to target morph to the original 2E9W dimer position, cached animation.
https://elifesciences.org/articles/64047#A5video1

## Appendix 6

## Mechanism-of-action animations for the insulin receptor

- The insulin receptor is a dimer of heterodimers that comprises two α-chains and two β-chains, represented as (αβ)2 (*Gutmann et al., 2018*).
- There are two insulin molecules per dimer (*Appendix 6—figure 1*; upper panel). Each is bound between L1 of one monomer and FnIII-1 of the other monomer. The inactive dimer is an inverted U shape.
- Insulin binding leads to a conformational change from the inactive U-shaped dimer to the active T-shaped dimer (*Appendix 6—figure 1*; lower panel).

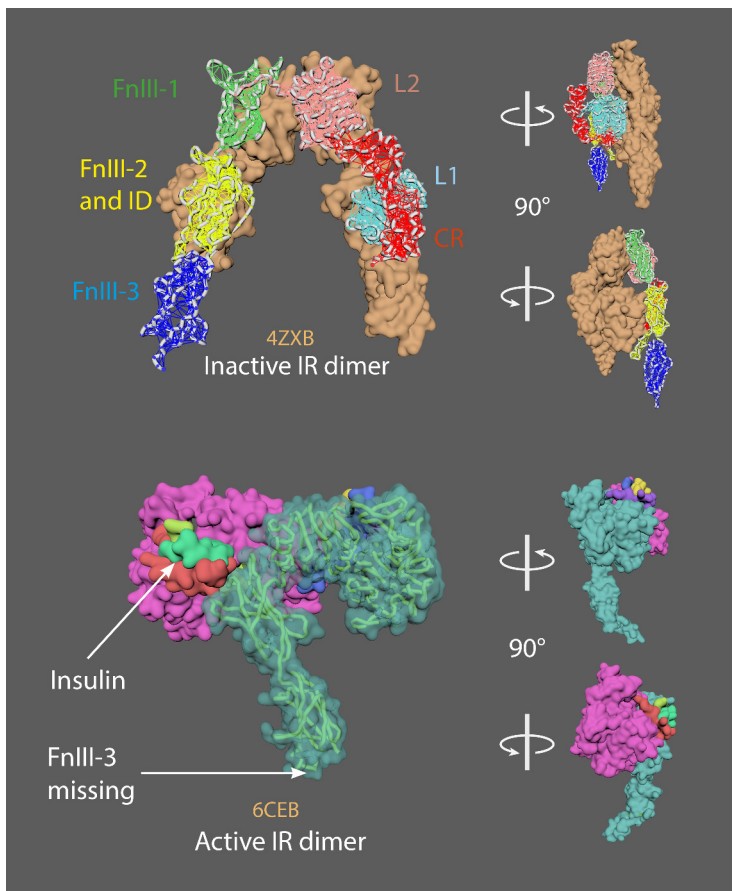

**Appendix 6—figure 1.** The inactive insulin receptor dimer has an inverted U shape (upper panel). The active insulin receptor dimer has a T shape (lower panel); the insulin ligand is shown in green.

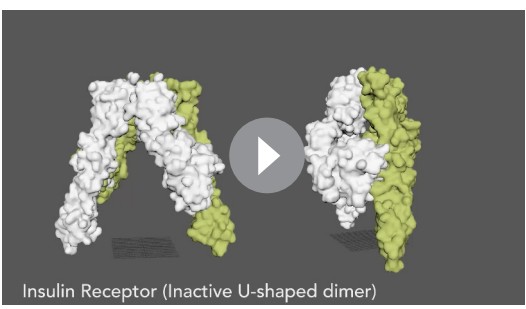

**Appendix 6—video 1.** Animation strategy for the insulin receptor.     4ZXB = inactive model (–ligands; *Croll et al., 2016*). 6CEB = active model (+ligands; *Scapin et al., 2018*). The missing FnIII-3 domain in 6CEB chain A was added using the mMaya modelling kit to get the whole ectodomain structure (FnIII-3 domain taken from 4ZXB); a new PDB was created. A mMaya rig was made for 4ZXB and elastic networks created for L1, CR, L2, FnIII-1, FnIII-2 and ID, and FnIII-3 domains. Rig was target morphed to the new version of 6CEB, mesh animation was cached. Binding animation of insulin ligands on 4ZXB were key-framed manually and was approximated based on their position relative to 6CEB.

https://elifesciences.org/articles/64047#A6video1

# Appendix 7

## Mechanism-of-action animations for the surface receptor Tetraspanin CD81 (TAPA-1)

- The structure of Tetraspanin CD81 resembles a waffle cone when bound with cholesterol (*Appendix 7—figure 1*).
- Cholesterol binding regulates CD81-mediated export of CD19.
- EC2 extracellular domain covers an intramembrane cavity of 4 transmembrane helices (TM 1–4).
- In the absence of cholesterol, EC2 adopts an ''open'' conformation.
- A salt bridge between EC2 and TM4 (D196–K201) stabilises the closed conformation and breaks during opening.
- Another salt bridge (K116–D117) forms upon opening which stabilises the open conformation.

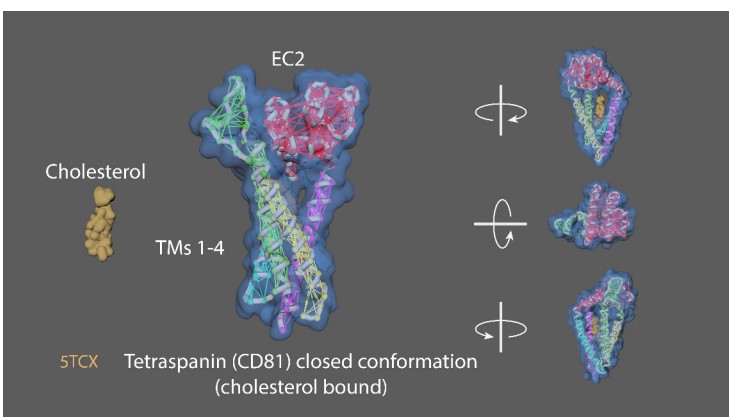

**Appendix 7—figure 1.** The structure of Tetraspanin CD81.

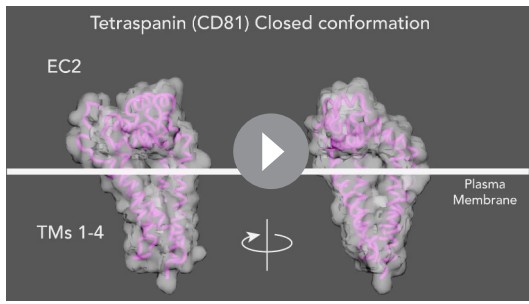

**Appendix 7—video 1.** Animation strategy for Tetraspanin. 5TCX = Tetraspanin (closed conformation (i.e. cholesterol-bound); *Zimmerman et al., 2016*). Rigged 5TCX, created elastic networks for EC2 (residues 34–55 and 113–201), TM1 (6–33), TM2 (56–84), TM3 (86–112), TM4 (202–232). Created a handle for the residues 113–201 of the EC2 extracellular domain, manually key-framed to form the open conformation (apoprotein/unbound to cholesterol).
https://elifesciences.org/articles/64047#A7video1

## Appendix 8

### Mechanism-of-action animations for the TNFR superfamily of surface receptors

The TNFR superfamily contains three groups of receptors: (i) death receptors (DRs); (ii) TNFR-associated factor (TRAF)–interacting receptors; (iii) decoy receptors (DcRs) (*Naismith et al., 1996*; *Vanamee and Faustman, 2018*)

- *Appendix 8—figure 1* shows several views of the DcR3-FASL complex, consisting of a trimeric ligand (green) bound to three upright decoy receptors (orange/purple).
- *Appendix 8—figure 2* shows the top and side views of a TNFSF protein complex featuring DR5 ligand (purple), TRAIL (blue) and antibody Fab fragments (orange/yellow) which was used to guide the hexagonal spacing in the representative TFNR1 array.
- The basic unit of signalling is a trimeric ligand and three receptors.
- Each receptor binds on the outside interface of two ligand monomers
- Receptors can pre-assemble (as resting or 'non-signalling' state) on the cell surface and can form either parallel or antiparallel dimers.
- Preferred model is antiparallel dimers.
- Antiparallel dimers are formed between the receptor CRD1 and CRD2 domains which occludes the ligand-binding site.
- Arranged as a large hexagonal lattice where each point connects three receptor monomers.
- Individual ligand-receptor complexes ~ 120 Å apart. Total edge length of ~170 Å (may vary with receptor type).
- Ligand binding leads to a conformational change in receptors (into an upright position, perpendicular to the cell surface) allowing cell signalling but maintaining hexagonal symmetry.

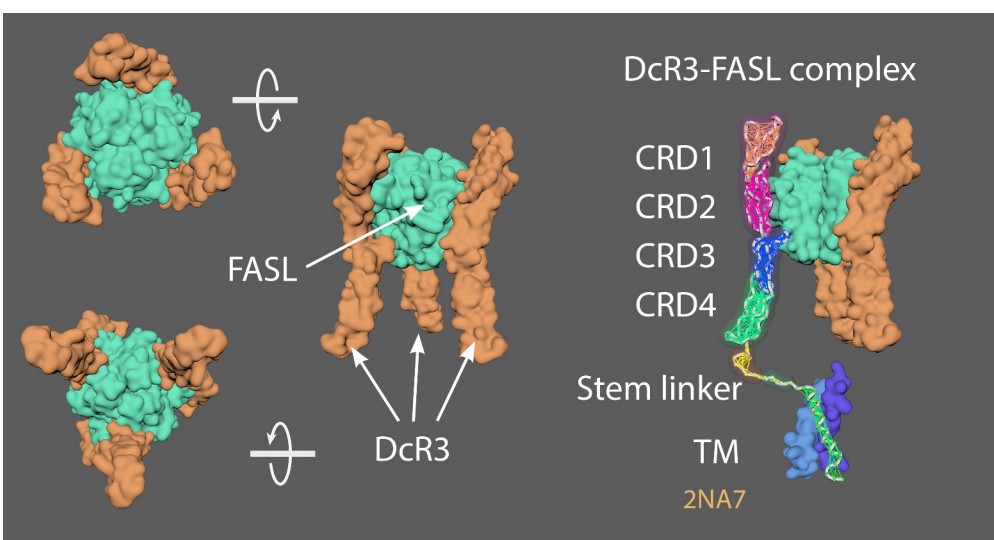

**Appendix 8—figure 1.** Different views of the DcR3-FASL complex, which consists of a trimeric ligand (green) bound to three upright decoy receptors (orange/purple).

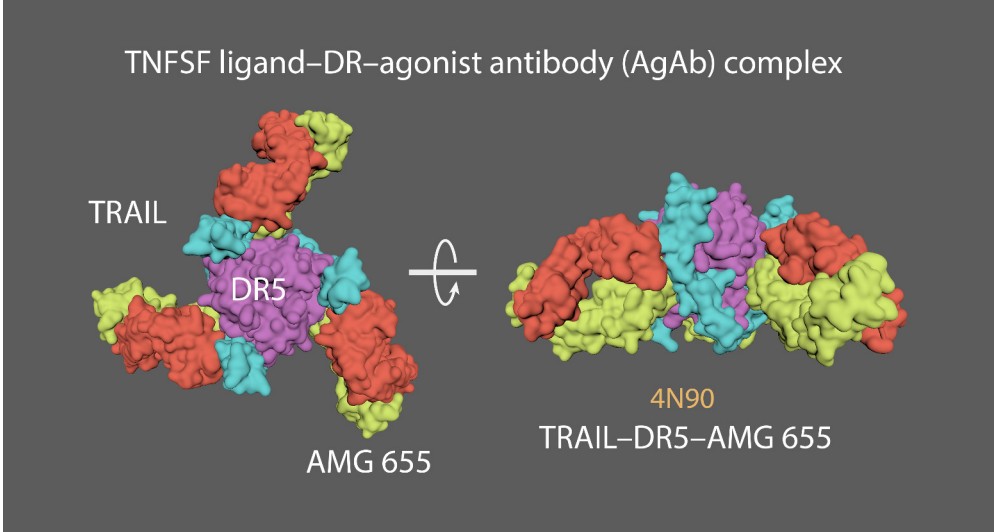

**Appendix 8—figure 2.** Top and side views of a TNFSF protein complex featuring DR5 ligand (purple), TRAIL (blue) and antibody Fab fragments (orange/yellow).

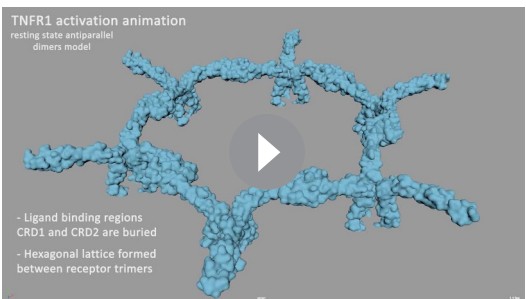

**Appendix 8—video 1.** Animation strategy for TNFR1.  4MSV = Decoy receptor 3 (DcR3)(chain A) and FasL (chain B; *Liu et al., 2016*). 2NA7 = Transmembrane domain of human Fas/CD95 death receptor (*Fu et al., 2016*). 4N90 = Crystal structure of TRAIL-DR5 with the agonist antibody (*Graves et al., 2014*). As no complete structures of the active state or resting/non-signalling state structures exist, a 'hybrid' TNFRSF molecule was made. Created a stem linker (PQIEN VKGTE DSGTT) taken from TNFR1 (1EXT4, residues 197–211) with mMaya modelling kit to connect the CRD4 of DcR3 (4 MSV) and a TM domain trimer (2NA7). New PDB created named 'Hybrid TNFRSF' and rigged with mMaya rigging kit. Elastic networks of CRD1-4, stem linker and TM domain were created. Handles were created (CD1 +2; CD3, and CD4) to manually move the rig into a hypothetical resting state. Movement was key-framed, animation cached and meshes extracted. Alembic cached meshes were positioned in a hexagonal lattice and ligand binding was key-framed for the final animation. 4N90 [Crystal structure of TRAIL-DR5 with the agonist antibody (Fab fragments)] was used as a reference to get the correct spacing of the hexagonal array when the trimers become active (from their antiparallel dimer conformation in the resting state).

https://elifesciences.org/articles/64047#A8video1

## Appendix 9

## Mechanism-of-action animations for the GLUT1 surface receptor

- GLUT1 (glucose transporter 1) is over-expressed in many cancer cells.
- 12 transmembrane (TM) segments form the N-terminal and C-terminal domains (*Appendix 9—figure 1*; centre)
- Rotation of the N-terminal and C-terminal domains occurs to make the transition from an outward-open state (*Appendix 9—figure 1*; lower left) to an outward-occluded state and, finally, to an inward-open state (*Appendix 9—figure 1*; upper left) to allow D-glucose transport.

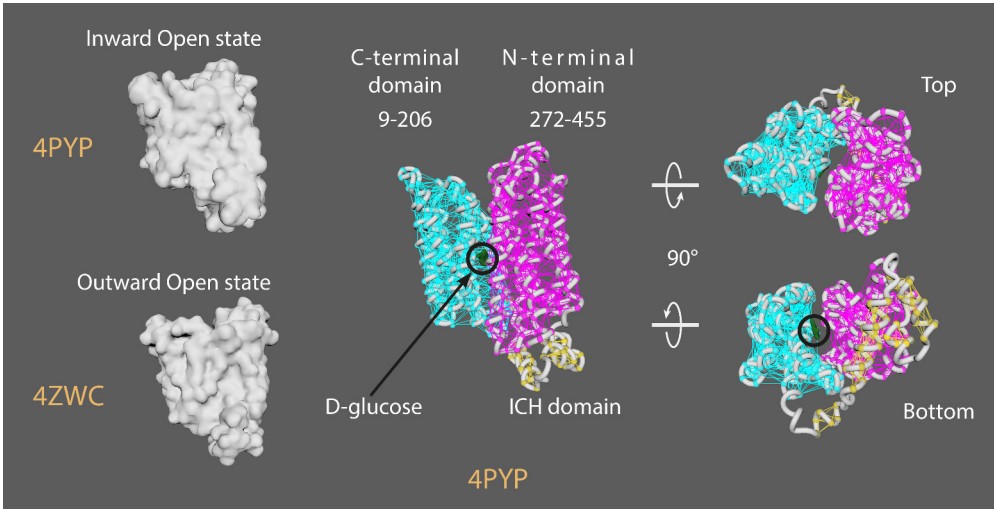

**Appendix 9—figure 1.** Different views of the GLUT1 surface receptor, including its outward-open and inward-open states.

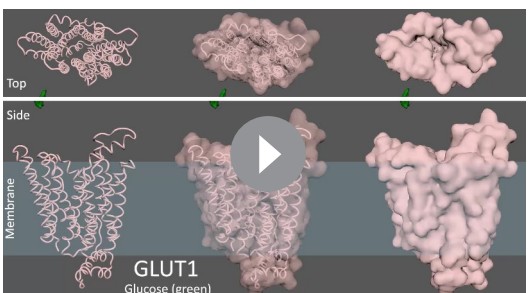

**Appendix 9—video 1.** Animation strategy for GLUT1.    4PYP = GLUT1; inward-open state (*Deng et al., 2015*). 4ZWC = GLUT3; outward-open state (*Deng et al., 2014*). Rigged 4PYP (chain A) (inward-open state) was targeted to conformationally morph 'backwards' into 4ZWC (outward-open state), and the alembic cache was reversed. Created elastic networks of N-terminal domain (9-206), C-terminal domain (272-455), and ICH domain (211-264). Downloaded Het atoms for D-glucose structure and manually key-framed glucose uptake.
https://elifesciences.org/articles/64047#A9video1

