## [Decision Letter]

Thank you for submitting your article "Meta-Research: Nanoscape, a data-driven 3D real-time interactive virtual cell environment" to *eLife* for consideration as a Feature Article. Your article has been reviewed by three peer reviewers, and the evaluation has been overseen by two members of the *eLife* Features Team (Helga Groll and Peter Rodgers). The following individuals involved in review of your submission have agreed to reveal their identity: Gael McGill (Reviewer #1); Jodie Jenkinson (Reviewer #2).

The reviewers were mostly positive about the article, but they raised a number of concerns, so we are inviting you to submit a revised version of the manuscript that addresses these comments.

*Reviewer #1:*

Nanoscape, a data-driven 3D real-time interactive virtual cell environment by Kadir et al. describes the creation of an advanced molecular and cellular landscape that can be explored interactively by users. The authors describe the numerous technical and design considerations and challenges inherent to such an endeavor and provide detailed information on both the software and workflow processes they employed to meet these challenges.

A clear strength of the paper is not only the multidisciplinary approach that was used to tackle such a complex visualization, but also the extensive technical expertise brought to bear on this task. Whereas many biomedical animators or teams of animators often rely on a relatively limited (yet advanced) software solution to create their work, McGhee et al's team explored the use of many advanced solutions and deployed them where most relevant. This team combined software ranging from 'Hollywood-style' animation packages (Maya, Houdini, Blender), specialized plugins that allow not only structural and other scientific data to be brought in to these 3D packages but also to create guided/coarse-grained conformational morphs and procedural environment creation (BioBlender, Molecular Maya, autoPack), molecular graphics packages commonly used in the scientific community (Chimera), molecular dynamics simulations (CHARMM), and finally game engine technology (Unity). This is an impressive array of software solutions to deploy and the authors do so with skill.

Despite the technical prowess and impressive visual result, one unfortunate weakness of the paper as currently written is the lack of a clear definition of the 'target user' and a more explicit statement of the intended learning objectives (or intended impact on the user). While the authors do a nice job of addressing the myriad limitations intrinsic to attempting an 'authentic' visualization of molecular and cellular worlds, they do not spend much time describing the design parameters that actually drove their decision-making process. This is touched upon in the 'Nanoscape User Experience' section, but in a rather fleeting way where much of the rich and detailed work and technical description provided in previous sections is rather abruptly wiped away with statements like "depiction of extreme molecular crowding and the ECM had to be compromised." While the decision itself is absolutely defendable, the reasoning behind it is not properly fleshed out. Given the technical expertise of the team, why not address such limitations by giving the user control of crowding parameters? This is offered as a potential next step in the 'Conclusion', but seems fundamental to a study that explores the difficult balance between what to show and what not to show. As noted above, this reviewer feels that this is partly due to the lack of a clear definition of what the authors consider a "user." Should the reader assume that this is a non-scientist – and if so, a student? At what educational level? Or is this a science museum goer? What are the specific goals of this experience? The authors mention wanting to promote "some cell biology fundamentals," but which ones specifically? Why is this set of visual experiences centered around breast cancer?

A more specific definition of 'user' would also help justify some of the important design decisions that were made both in terms of the representational choices (coarser protein meshes processed in Zbrush, and simplified lipid representations with a texture instead of meshes) but also the simplifications in dynamics of molecular actors (where the authors describe a process of developing 'MoA' animations based on coarse-grained simulations, but then using these only to inform more traditional 'rigging and animation' workflow typical of the entertainment industry). I would also have liked to read more specifics about how the different levels and environments in the visualization were connected together – how does the viewer navigate from one environment to another? Does this experience contains UI/interactive elements to do this beyond the freedom to gaze in any direction?

Finally, the authors do not touch at all on a discussion of how they would test the efficacy of their visualization on their target audience. While including such a protocol and carrying such work may not be part of this study and beyond the scope of the authors' initial project, I would recommend that they at least address how they would go about such an endeavor because the design challenges they describe and acknowledge throughout the paper cannot, ultimately, be 'guessed' – they have to be tested empirically.

Overall, the authors have accomplished an impressive amount of work and created what is no doubt a stunning experience. This is also evident in the quality of the images and figure design in the manuscript itself. The supplementary tables and figures are highly detailed and very useful to any who would seek to understand and recreate some of the technical aspects of this work. Although the authors discuss and lament the lack of a database beyond the PDB to share and annotate integrative modeling efforts like theirs, they do not follow-up by offering to share the models created in this study as a way to remedy such a problem. Nevertheless, I would recommend that this study be published in *eLife* as I believe it represents an impressive effort towards the advanced visualization of environments relevant to the readership. I would however ask that the authors address the points made above and summarized here for clarity here:

1. Please provide a clearer definition or discussion of your intended user.

2. For this user, what are the specific learning/communication objectives?

3. Draw more direct connections between these objectives and the specific design decisions that drove your project implementation.

4. Although carrying out an assessment study may be beyond the scope of this manuscript, please discuss how this could be carried out and, in doing so, provide a more explicit statement of learning goals and outcomes.

5. I would also want to know (and have the authors describe) whether they intend to share their models and the experience itself (since they say it can be set up on a user desktop – as opposed to their previous project which appears to require a more involved set-up for VR).

6. Finally, I find the suggestion in the conclusion that such environments may become useful for scientists to 'reflect upon their own data' is very interesting. However, such a statement requires some level of discussion addressing how visual design choices may need to change to accommodate the goals of this new kind of audience. Is there a difference in what molecular actors (or representational choices) could be left out or included, and what specific aspects of this interactive and immersive environment do the authors find most promising for use by the scientific community?

Reviewer #2:

This manuscript reports on the design considerations undertaken in the development of virtual reality cell explorer "Nanoscape". The authors describe in detail the process by which molecular entities and extracellular elements are sourced, modelled, distributed, and ultimately rendered. Detailed rationale is provided for the stylistic decisions both as it pertains to technological concerns and clarity of representation. For researchers working in the domain of molecular visualization this offers an interesting and insightful overview of this collaborative project.

I thoroughly enjoyed reviewing this manuscript, particularly because of the high level of detail and transparency provided by the authors with respect to the ways in which the environment they have created has been informed by empirical data. They describe the ways in which data has been modified and in each case provide rationale for decisions made (modifications or simplification of data as well as the use of artistic license for visual clarity). The supplemental materials too were very informative. This manuscript represents a standard for how all projects of this type should be documented and reported.

I have but one concern that is relatively very minor. It would be helpful if the authors provided some additional context as to who their users are? The users are mentioned several times throughout the manuscript as a consideration in design decisions made, but no clear characterization of the anticipated user group is ever offered.

In sum, I think this manuscript makes a valuable contribution to the domain of molecular visualization from both a technical and design perspective.

*Reviewer #3:*

This article discusses the use of computer visualization techniques to create an explorable 3D virtual environment, called Nanoscape, depicting a breast cancer cell within the context of a tumor.

1. Although the title suggests that the article will describe the Nanoscape application in detail, we were disappointed that only a cursory description of this project was given. While we understand that Nanoscape has yet to be released (and so user feedback should not be expected), we hoped to better understand what the purpose of the overall project is. Who is the target audience, how would the application be implemented, and what do the authors hope to achieve? An explanation of the novel features of Nanoscape (in comparison to other similar efforts) is particularly important given that many of the software techniques that have been used for this project have been previously released and/or described by others (e.g.Molecular Maya for protein modeling, CellPack for creating crowded environments) and do not represent new methods.

2. We were concerned that the article appears to be written primarily for 3D biomedical illustrators, rather than for research biologists. The article itself questioned whether 3D visualizations could support experimentalists (see lines 115-117). Given the readership of *eLife*, we felt that a much stronger consideration for the role of 3D visualization in the research sphere, and/or a more in-depth discussion of how these types of applications (and Nanoscape in particular) can impact public engagement and outreach is warranted.

3. In many sections of the article, the authors discussed general challenges faced when creating molecular environments, but did not discuss how they met these challenges in creating the Nanoscape models and how they made the decisions that they ultimately made. For example, the authors discussed the issue of conformational flexibility of proteins (paragraph starting on line 227) and the limitation posed when only one conformation of a protein has been structurally solved. How did the authors solve this problem in the case of Nanoscape? Likewise, when discussing the dynamics of lipids within a membrane, the authors point out that depicting the movement and heterogeneity of membranes is challenging, but did not describe how this challenge was met in the context of Nanoscape.

4. In line 202, mMaya is described as a software that "qualitatively replicate(s) molecular dynamics," and this software is described throughout the article as the main tool that was used to create dynamic animations of different proteins. A direct comparison of the process and computed trajectories produced by mMaya and those produced by established molecular dynamics simulation packages and/or coarse-grained modeling would ideally be provided to better support this description of mMaya, especially considering that mMaya is likely to be a software that is unfamiliar to many readers, and that there are (as far as we know) no other publications that describe the use of mMaya in the scientific literature.

---

## [Author Response]

Reviewer #1:[…]Overall, the authors have accomplished an impressive amount of work and created what is no doubt a stunning experience. This is also evident in the quality of the images and figure design in the manuscript itself. The supplementary tables and figures are highly detailed and very useful to any who would seek to understand and recreate some of the technical aspects of this work. Although the authors discuss and lament the lack of a database beyond the PDB to share and annotate integrative modeling efforts like theirs, they do not follow-up by offering to share the models created in this study as a way to remedy such a problem. Nevertheless, I would recommend that this study be published in eLife as I believe it represents an impressive effort towards the advanced visualization of environments relevant to the readership. I would however ask that the authors address the points made above and summarized here for clarity here:1. Please provide a clearer definition or discussion of your intended user.

We more clearly define the primary user group as tertiary and university biology students. Secondary users have also been identified (public, experimental scientists), but are not the core target group.

2. For this user, what are the specific learning/communication objectives?

We more clearly define that the key objective of Nanoscape in this group is to support the learning of advanced biological concepts such as scale, density and interactions which may be misleading or oversimplified in other learning material. New Section in paper: Nanoscape Evaluation and broader applications

3. Draw more direct connections between these objectives and the specific design decisions that drove your project implementation.

Design decisions were paramount for both computational performance and aesthetic comfort. To avoid being visually overwhelmed, we had to reduce the number of proteins and small molecules (eg. water) in the environment. This also allowed a better appreciation of the 'vistas' of organic landscape and how several components interact with one another. In another example, the complexity of the ECM had to be simplified due to uncertainties about how the components co-ordinate their interactions with surface receptors. Proteins were stylised as polygonal meshes to highlight protein domains (some later animated), instead of atomic representation. In most cases, the appearance of the mesh of each component (protein/cell/blood vessel, etc.) was modified (made low-poly) to ensure real-time performance.

4. Although carrying out an assessment study may be beyond the scope of this manuscript, please discuss how this could be carried out and, in doing so, provide a more explicit statement of learning goals and outcomes.

We suggest that evaluation of the application for measures of usability (attention, cognitive demand etc) and educational value (as a didactic aid) is vital and is planned for phase 2 of this project. We would like to understand if Nanoscape, with its increased level of authenticity can help deepen the understanding of advanced topics such as protein density and molecular scale. Assessors can vary these parameters in application and evaluate the effect each has on basic and advanced cellular topics through exams. We would also like to test the effect of immersion and interactivity of this application against a more linear version of the content (like a pre-recorded video).

5. I would also want to know (and have the authors describe) whether they intend to share their models and the experience itself (since they say it can be set up on a user desktop – as opposed to their previous project which appears to require a more involved set-up for VR).

We will be making this publicly accessible (likely via the Steam store).

6. Finally, I find the suggestion in the conclusion that such environments may become useful for scientists to 'reflect upon their own data' is very interesting. However, such a statement requires some level of discussion addressing how visual design choices may need to change to accommodate the goals of this new kind of audience. Is there a difference in what molecular actors (or representational choices) could be left out or included, and what specific aspects of this interactive and immersive environment do the authors find most promising for use by the scientific community?

Have touched on the value to scientists – we believe that this application probably won’t offer a platform for rigorous data interrogation, but perhaps more holistic reflection of how systems integrate by seeing several modalities of data presented in the one cohesive format. Users may also gain an appreciation of the piece simply as an aesthetic piece of art or entertainment.

Reviewer #2:[…]I have but one concern that is relatively very minor. It would be helpful if the authors provided some additional context as to who their users are? The users are mentioned several times throughout the manuscript as a consideration in design decisions made, but no clear characterization of the anticipated user group is ever offered.

Please see reviewer 1, comments 1-2.

Reviewer #3:This article discusses the use of computer visualization techniques to create an explorable 3D virtual environment, called Nanoscape, depicting a breast cancer cell within the context of a tumor.1. Although the title suggests that the article will describe the Nanoscape application in detail, we were disappointed that only a cursory description of this project was given. While we understand that Nanoscape has yet to be released (and so user feedback should not be expected), we hoped to better understand what the purpose of the overall project is. Who is the target audience, how would the application be implemented, and what do the authors hope to achieve? An explanation of the novel features of Nanoscape (in comparison to other similar efforts) is particularly important given that many of the software techniques that have been used for this project have been previously released and/or described by others (e.g.Molecular Maya for protein modeling, CellPack for creating crowded environments) and do not represent new methods.

Please see reviewer 1, comments 1-2.

2. We were concerned that the article appears to be written primarily for 3D biomedical illustrators, rather than for research biologists. The article itself questioned whether 3D visualizations could support experimentalists (see lines 115-117). Given the readership of eLife, we felt that a much stronger consideration for the role of 3D visualization in the research sphere, and/or a more in-depth discussion of how these types of applications (and Nanoscape in particular) can impact public engagement and outreach is warranted.

Please see reviewer 1, comment 6.

3. In many sections of the article, the authors discussed general challenges faced when creating molecular environments, but did not discuss how they met these challenges in creating the Nanoscape models and how they made the decisions that they ultimately made. For example, the authors discussed the issue of conformational flexibility of proteins (paragraph starting on line 227) and the limitation posed when only one conformation of a protein has been structurally solved. How did the authors solve this problem in the case of Nanoscape? Likewise, when discussing the dynamics of lipids within a membrane, the authors point out that depicting the movement and heterogeneity of membranes is challenging, but did not describe how this challenge was met in the context of Nanoscape.

In many cases, proteins existed only in one conformational state, so molecular animations were not possible for those. This application did not intend to offer rigorous molecular dynamics simulations, as this was outside the scope of the work. Where several protein states existed, we were able to animate those proteins with a reasonable level of confidence. Another option that we did not explore was to simply use artistic licence and artistically manipulate conformational changes to suggest flexibility of the proteins, but this was not completed for the current version of the application. We also note that due to performance reasons, we elected to simulate the cell membrane as an animated texture instead of individual lipids.

4. In line 202, mMaya is described as a software that "qualitatively replicate(s) molecular dynamics," and this software is described throughout the article as the main tool that was used to create dynamic animations of different proteins. A direct comparison of the process and computed trajectories produced by mMaya and those produced by established molecular dynamics simulation packages and/or coarse-grained modeling would ideally be provided to better support this description of mMaya, especially considering that mMaya is likely to be a software that is unfamiliar to many readers, and that there are (as far as we know) no other publications that describe the use of mMaya in the scientific literature.

Have included a detailed description of mMaya from its publishers in the supplementary figures.